# Auction-Based Regulation for Artificial Intelligence

## Abstract

In an era of "moving fast and breaking things", regulators have moved slowly to pick up the safety, bias, and legal pieces left in the wake of broken Artificial Intelligence (AI) deployment. Since AI models, such as large language models, are able to push misinformation and stoke division within our society, it is imperative for regulators to employ a framework that mitigates these dangers and ensures user safety. While there is much-warranted discussion about how to address the safety, bias, and legal woes of state-of-the-art AI models, the number of rigorous and realistic mathematical frameworks to regulate AI safety is lacking. We take on this challenge, proposing an auction-based regulatory mechanism that provably incentivizes model-building agents (i) to deploy safer models and (ii) to participate in the regulation process. We provably guarantee, via derived Nash Equilibria, that each participating agent's best strategy is to submit a model safer than a prescribed minimum-safety threshold. Empirical results show that our regulatory auction boosts safety and participation rates by $20\%$ and $15\%$ respectively, outperforming simple regulatory frameworks that merely enforce minimum safety standards.

## 1 Introduction

Current Artificial Intelligence (AI) models are powerful and have revolutionized a wide swath of industries. The recent large-scale deployment of Large Language Models (LLMs) has simultaneously boosted human productivity while sparking concern over safety (*e.g.,* hallucinations, bias, and privacy). We have seen industry leaders, such as Google, Meta, and OpenAI, embroiled in controversy surrounding bias and misinformation (Brewster, 2024; Robertson, 2024; White, 2024), safety (Jacob, 2024; Seetharaman, 2024; White, 2023), as well as legality and ethics (Bruell, 2023; Metz et al., 2024; Moreno, 2023) in their development and deployment of LLMs. Furthermore, irresponsible deployment of LLMs runs the risk of allowing adversaries the ability to spread misinformation or propaganda (Barman et al., 2024; Neumann et al., 2024; Sun et al., 2024). Unfortunately, a consistent and industry-wide solution to oversee safe AI deployment remains elusive.

Naturally, one such solution to mitigate these dangers is for increased governmental regulation over AI deployment. In the United States, there have been some strides, on federal (House, 2023) and state levels (Information, 2024), to regulate the safety and security of large-scale AI systems (including LLMs). While these recent executive orders and bills highlight the necessity to develop safety standards and enact safety and security protocols, few details are offered. This follows a consistent trend of well-deserved scrutiny towards the lack of AI regulation without the development of rigorous and realistic mathematical frameworks to regulate. Our work sets out to solve this disconnect by proposing a novel regulatory framework that a regulator can follow to not only strictly enforce the safety of deployed AI models, but simultaneously *incentivize the production of safer AI models.*

The goal of our work is to formulate the AI regulatory process mathematically, and subsequently develop a mechanism to incentivize safer development and deployment of AI models. Specifically, we formulate the AI regulatory process as an *all-pay auction*, where agents (companies) submit their models to a regulator. The regulator's job is twofold: **(a)** prohibit deployment of models that fail to meet prescribed safety thresholds, and **(b)** incentivize safe model production and deployment

Figure 1: **Step-by-Step SIRA Schematic.** (Step 0) The regulator sets a safety threshold, $\epsilon$, having a corresponding price, $p_\epsilon$, required to achieve $\epsilon$. (Step 1) Agents evaluate their total value, $V_i$, from model deployment value ($v_i^d$) and potential regulator compensation ($v_i^p$). Agents only participate if their total value exceeds $p_\epsilon$. (Step 2) Participating agents submit their models to the regulator, accompanied by their bid $b_i$, which reflects the amount spent to improve their model's safety level. Models with bids below $p_\epsilon$ are automatically rejected. (Step 3) The submitted models are randomly paired, and the safer model (i.e., the higher bid) in each pair wins the comparison. In this example, agent 3 wins since $b_3 > b_1$. (Step 4) Winning models receive both a premium and deployment value (i.e., agent 3 in this example wins premium $v_3^p$ and deployment value $v_3^d$), while losing models receive only the deployment value (i.e., agent 1 in this example wins deployment value $v_1^d$ only).

by providing additional rewards to agents that submit safer models than their peers. We design an auction-based regulatory mechanism, with a novel reward-payment protocol, that emits Nash Equilibria at which agents develop and deploy models safer than the prescribed safety threshold.

**Summary of Contributions**. In summary, the main contributions of our paper are as follows.

- **(AI Regulation):** We propose a Safety-Incentivized Regulatory Auction (SIRA) mechanism, offering a simple yet practical approach to AI regulation.
- **(Safety-First):** We establish, through derived Nash Equilibria, that agents are incentivized to submit models that surpass the required safety threshold.
- **(Effective):** Empirical results demonstrate that SIRA increases model safety by over 20% and boosts participation rates by 15% compared to baseline regulatory mechanisms.

## 2 RELATED WORKS

**Regulation Frameworks for Artificial Intelligence.** A handful of work focuses on regulation frameworks for AI (de Almeida et al., 2021; Jagadeesan et al., 2024; Rodríguez et al., 2022; Yaghini et al., 2024). First, de Almeida et al. (2021) details the need for AI regulation and surveys existing proposals. The proposals are ethical frameworks detailing specific ethical decisions to make and dilemmas to address. These proposals lack a mathematical framework to incentivize provably safer models. Rodríguez et al. (2022) utilize AI models to detect collusive auctions. This work is related to our own but in reverse: AI is applied to regulate auctions and ensure that they are not collusive. In contrast, our work aims to use auctions to regulate AI deployment. Jagadeesan et al. (2024) focuses on reducing barriers to entry for smaller companies who are competing against larger and more established incumbent companies. A multi-objective high-dimensional regression framework is proposed to capture "reputational damage" for companies who deploy unsafe AI models. This work allows varying levels of safety constraints, where newer companies face less severe constraints in order to spur their entry into the market, which is unrealistic in many settings and only considers simple linear-regression models. The closest related work to ours, Yaghini et al. (2024), proposes

a regulation game for ensuring privacy and fairness that is formulated as a Stackelberg game. This game is a multi-agent optimization problem that is also multi-objective (for fairness and privacy). An equilibrium-search algorithm is presented to ensure that agents remain on the Pareto frontier of their objectives (although this frontier is estimated algorithmically). Notably, Yaghini et al. (2024) considers only one model builder (agent) and multiple regulators that provide updates to the agent's strategy. We consider a more realistic setup, where there are multiple agents and a single regulator whose goal is to incentivize safer model deployment. We do not believe that it is the regulator's job to collaborate with agents to optimize their strategy. Furthermore, our mechanism is simple and efficient; we do not require Pareto frontier estimation or multiple rounds of optimization.

**All-Pay Auctions.** Compared to the dearth of literature in regulatory frameworks for AI, all-pay auctions are well-researched (Amann & Leininger, 1996; Baye et al., 1996; Bhaskar, 2018; DiPalantino & Vojnovic, 2009; Gemp et al., 2022; Goeree & Turner, 2000; Siegel, 2009; Tardos, 2017). These works formulate specific all-pay auctions and determine their equilibria. Some works consider settings where agents have complete information about their rivals' bids (Baye et al., 1996) while others consider incomplete information, such as only knowing the distribution of agent valuations (Amann & Leininger, 1996; Bhaskar, 2018; Tardos, 2017). One major application of all-pay auctions are crowd-sourcing competitions. Many agents participate to win a reward, with those losing still incurring a cost for their time, effort, *etc*. DiPalantino & Vojnovic (2009) is one of the first works to model crowd-sorucing competitions as an all-pay auction. Further research, such as Gemp et al. (2022), have leveraged AI to design all-pay auctions for crowd-sourcing competitions. Instead of crowd-sourcing, our work formulates the AI regulatory process as an asymmetric and incomplete all-pay auction. We leverage previous analysis in this setting (Amann & Leininger, 1996; Bhaskar, 2018; Tardos, 2017) to derive equilibria.

## 3 THE REGULATORY AI SETTING

There exists a regulator $R$ with the power to set and enforce laws and regulations (*e.g.,* U.S. government regulation on lead exposure). The regulator wants to regulate AI model deployment, by ensuring that all models meet a given safety threshold $\epsilon \in (0, 1)$, *e.g.,* the National Institute for Occupational Safety and Health regulates that N95 respirators filter out at least 95% of airborne particles. If a model does not reach the safety criteria $\epsilon$, then the model is deemed unsafe and the regulator bars deployment. On the other side, there are $n$ rational model-building agents. Agents seek to maximize their own benefit, or utility.

**Bidding & Evaluation**. By law, each agent $i$ must submit, or bid in auction terminology, its model $w_i \in \mathbb{R}^d$ for evaluation to the regulator before it can be approved for deployment. Let $S(w; x) : \mathbb{R}^d \to \mathbb{R}_+$ be a safety metric that outputs a safety level (the larger the better) for model $w$ given data $x$. In effect, each agent, given its own data $x_i$, bids a safety level $s_i^A := S(w_i; x_i)$ to the regulator. Subsequently, the regulator, using its own data $x_R$, independently evaluates the agent's safety level bid as $s_i^R := S(w_i; x_R)$. We assume that agent and regulator evaluation data is independent and identically distributed (IID) $x_i, x_R \sim \mathcal{D}$.

**Assumption 1.** *Agent and regulator evaluation data comes from the same distribution $x_i, x_R \sim \mathcal{D}$.*

This assumption is often realistic in regulatory settings, because both agents and regulators typically rely on standardized or widely accepted datasets for model evaluation, ensuring a fair and unbiased assessment of safety levels. For instance, when assessing LLMs, common datasets like benchmarks for toxicity or bias are employed consistently across evaluations, reflecting real-world data distributions. Therefore, it is reasonable to define agent $i$'s safety level bid as $s_i := \mathbb{E}_{x \sim \mathcal{D}}[S(w_i; x)]$. We address scenarios where evaluation data may not follow the IID assumption in Section 7.

**The Price of Safety**. We assume that there exists a strictly increasing function $M : (0, 1) \to (0, 1)$ that determines the "price of safety" (*i.e.,* maps safety into cost). Simply put, safer models cost more to attain. As a result, we define the price of attaining $\epsilon$ safety as $p_\epsilon := M(\epsilon)$.

**Assumption 2.** *There exists a strictly increasing function $M$ that maps safety to cost.*

The assumption that a strictly increasing function $M$ maps safety to cost is realistic, because achieving higher safety levels typically requires greater resources. Safer models often demand more data, advanced tuning, and extensive validation, all of which increase costs. Thus, defining the price of safety as $p_\epsilon := M(\epsilon)$, where $M$ is strictly increasing, reflects the practical trade-off that safer models cost more to develop.

**Agent Costs**. Unfortunately for agents, training a safer model comes with added cost. Consequently, each agent $i$ must decide how much money to *bid*, or spend, $b_i$ to make its model safer. By Assumption 2, the resulting safety level of an agent's model will be $s_i = M^{-1}(b_i)$.

**Agent Values**. *(1) Model deployment value $v_i^d$*. While it costs more for agents to produce safer models, they gain value from having their models deployed. Intuitively, this can be viewed as the expected value $v_i^d$ of agent $i$'s model. The valuation for model deployment varies across agents (*e.g.,* Google may value having its model deployed more than Apple). *(2) Premium reward value $v_i^p$*. Beyond value for model deployment, the regulator can also offer additional, or premium, compensation valued as $v_i^p$ by agents (*e.g.,* tax credits for electric vehicle producers or Fast Track and Priority Review of important drugs by the U.S. Food & Drug Administration). The regulator provides additional compensation to agents whose models demonstrate safety levels exceeding the prescribed threshold. However, the value of this compensation varies across agents due to differing internal valuations. It is unrealistic for the regulator to compensate all agents meeting the safety threshold due to budget constraints. Therefore, we limit the additional rewards to a top-performing half of agents who surpass the threshold, ensuring that compensation targets those contributing the most to enhanced safety while maintaining feasibility for the regulator.

**Value Distribution**. We define the total value for each agent $i$ as $V_i := v_i^d + v_i^p$, which represents the sum of the deployment value and premium compensation. Although these values can vary widely in practice, we normalize $\{V_i\}_{i=1}^n$ for all $n$ agents to be between 0 and 1 for analytical tractability, allowing us to work within a standardized range. Consequently, the price to achieve the safety threshold $\epsilon$ is also normalized to fall within the $(0, 1)$ interval, i.e., $p_\epsilon \in (0, 1)$.

The proportion of total value allocated to deployment versus compensation is determined by a scaling factor $\lambda_i \sim \mathcal{D}_\lambda(0, 1/2)$. Therefore, the deployment value is $v_i^d := (1 - \lambda_i)V_i$, and the premium compensation value is $v_i^p := \lambda_i V_i$. Both $V_i$ and $\lambda_i$ are private to each agent, though the distributions $\mathcal{D}_V$ and $\mathcal{D}_\lambda$ are known by participants. We set the maximum allowable factor at $\lambda_i = 1/2$, reflecting the realistic constraint that compensation should not exceed deployment value. Although our results primarily consider $\lambda_i \leq 1/2$, theoretical extensions can be made for scenarios where $\lambda_i > 1/2$.

**All-Pay Auction Formulation**. Overall, agents face a trade-off: producing safer models garners value, via the regulator, but incurs larger costs. Furthermore, in order to attain the rewards detailed in Section 3, agents must submit a model with safety level at least as large as $\epsilon$. We can formulate this problem as an *asymmetric all-pay auction* with *incomplete information* (Amann & Leininger, 1996; Bhaskar, 2018; Tardos, 2017). The problem is an all-pay auction since agents incur an unrecoverable cost, safety training costs, when submitting their model to regulators. The problem is asymmetric with incomplete information since valuations $V_i$ are private and differ for each agent.

**Agent Objective**. The objective for each model-building agent $i$, is to maximize its own utility $u_i$. Namely, each agent seeks to determine an optimal safety level to bid to the regulator $b_i$. However, depending upon the all-pay auction formulation, agents would need to take into account all other agents' bids $\boldsymbol{b}_{-i}$ in order to determine their optimal bid $b_i^*$,

$$b_i^* := \arg\max_{b_i} u_i(b_i; \boldsymbol{b}_{-i}). \tag{1}$$

A major portion of our work is constructing an auction-based mechanism, thereby designing the utility of each agent, such that participating agents maximize their utility when they bid more than "the price to obtain the minimum safety threshold", i.e., $b_i^* > p_\epsilon$. We begin by providing a simple mechanism, already utilized by regulators, that does not accomplish this goal, before detailing our auction-based mechanism SIRA that provably ensures that $b_i^* > p_\epsilon$ for all agents.

## 4  RESERVE THRESHOLDING: BARE MINIMUM REGULATION

The simplest method to ensure model safety is for the regulators to set a reserve price, or minimum acceptable safety. We term this mechanism a *multi-winner reserve thresholding auction*, where the regulator awards a deployment reward, $v_i^d$, to each agent whose submitted model meets or exceeds the safety threshold $\epsilon$. Within this auction, each agent $i$'s utility is mathematically formulated as,

$$u_i(b_i; \boldsymbol{b}_{-i}) = \begin{cases} -b_i \text{ if } b_i < p_\epsilon, \\ v_i^d - b_i \text{ if } b_i \geq p_\epsilon. \end{cases} \tag{2}$$

The formulation above, however, is ineffective at incentivizing agents to produce models that are safer than the $\epsilon$ threshold.

**Theorem 1** (Reserve Thresholding Nash Equilibrium). *Under Assumption 2, agents participating in Reserve Thresholding* (2) *have an optimal bidding strategy and utility of,*

$$b_i^* = p_\epsilon, \quad u_i(b_i^*; \boldsymbol{b}_{-i}) = v_i^d - p_\epsilon, \tag{3}$$

*and submit models with the following safety level,*

$$s_i^* = \begin{cases} \epsilon & \text{if } u_i(b_i^*; \boldsymbol{b}_{-i}) > 0, \\ 0 \text{ (no model submission)} & \text{else.} \end{cases} \tag{4}$$

When a regulator implements reserve thresholding, as formally detailed in Theorem 1, agents exert minimal effort, submitting models that just meet the required safety threshold $\epsilon$. While this approach ensures that all deployed models satisfy the minimum safety requirements, it fails to encourage agents to build models with safety levels exceeding $\epsilon$. Additionally, agents whose deployment rewards are less than the cost of achieving the safety threshold, i.e., $v_i^d < p_\epsilon$, have no incentive to participate in the regulatory process, leading to reduced participation rates, as detailed in Remark 1.

**Remark 1** (Lack of Incentive). *Each agent is only incentivized to submit a model with safety $s_i^* = \epsilon$. Our goal is to construct a mechanism that incentivizes agents to build models which reach safety criteria greater than the minimum requirement: $s_i^* > \epsilon$.*

## 5  SAFETY-INCENTIVIZED REGULATORY AUCTIONS (SIRA)

To alleviate the lack of incentives within simple regulatory auctions, such as the one in Section 4, we propose a regulatory all-pay auction that emits an equilibrium where agents *submit models with safety levels larger than* $\epsilon$.

**Algorithm Description.** The core component of our auction is that agent safety levels are randomly compared against one another, with the regulator rewarding those having the safer model with premium compensation. Only agents with models that achieve a safety level of $\epsilon$ or higher are eligible to participate in the comparison process; models that do not meet this threshold are automatically rejected. The detailed algorithmic block of SIRA is depicted in Algorithm 1.

**Agent Utility.** The utility for each agent $i$ is therefore defined as in Equation (5).

$$u_i(b_i; \boldsymbol{b}_{-i}) = \left(v_i^d + v_i^p \cdot 1_{(\text{if } i \text{ wins comparison})}\right) \cdot 1_{(\text{if } b_i \geq p_\epsilon)} - b_i. \tag{5}$$

Per regulation guidelines, the safety criteria of an accepted model must at least be $\epsilon$. Equation (5) dictates that values are only realized by each agent if their model has a bid larger than the required cost to reach $\epsilon$ safety, $1_{(\text{if } b_i \geq p_\epsilon)}$. Furthermore, agents only realize additional compensation value $v_i^p$ from the regulator if their safety level outperforms a randomly selected agent, $1_{(\text{if } i \text{ wins comparison})}$. Any agent that bids $b_i = 1$ will automatically win and realize both $v_i^p$ and $v_j^w$. It is important to note that the cost that every agent incurs when building its model is sunk: if the model is not cleared for deployment, the cost $-b_i$ is still incurred. We rewrite the agent utility in a piece-wise manner below,

$$u_i(b_i; \boldsymbol{b}_{-i}) = \begin{cases} -b_i \text{ if } b_i < p_\epsilon, \\ v_i^d - b_i \text{ if } b_i \geq p_\epsilon \text{ and } b_i < b_j \text{ randomly sampled agent bid } b_j, \\ v_i^d + v_i^p - b_i \text{ if } b_i \geq p_\epsilon \text{ and } b_i > b_j \text{ randomly sampled agent bid } b_j. \end{cases} \tag{6}$$

---

**Algorithm 1** SIRA: Safety-Incentivized Regulatory Auction

---

**Require:** $n$ model-building agents and a safety level $\epsilon$ set by regulator $R$ (corresponding price $p_\epsilon$)
1: Each agent $i$ receives their total value $V_i$ and partition ratio $\lambda_i$ from "nature"
2: Agents determine their optimal bids $b_i^*$ and corresponding utility $u_i(b_i^*)$ ▷ via Corollaries 1 or 2
3: Agents decide to participate, the set of participating agents is $P = \{j \in [n] \mid u_j(b_j^*; \mathbf{b}_{-i}) > 0\}$
4: **for** participating agents $j \in P$ **do**
5:     Spend $b_j^*$ to build a model, with safety level $s_j = M^{-1}(b_j^*)$, and submit it to the regulator
6: Regulator verifies all model safety levels, clearing models for deployment when $s_j \geq \epsilon \; \forall j \in P$
7: Regulator pairs up models, awarding compensation valued at $v_i^p$ to agents with the safer model

---

By introducing additional compensation, $v_i^p$, and, crucially, conditioning it on whether an agent's model is safer than that of another random agent, we seek to make it rational for agents to bid more than the price to obtain the minimum safety threshold (unlike Theorem 1).

**Incentivizing Agents to Build Safer Models.** We establish a guarantee that agents participating in SIRA *maximize their utility with an optimal bid $b_i^*$ that is larger than "the price required to attain $\epsilon$ safety" (i.e., $b_i^* > p_\epsilon$)* in Theorem 2 below. Furthermore, agents bid in proportion to the value for additional compensation $v_i^p$ that the regulator offers for extra safe models.

**Theorem 2.** *Agents participating in* SIRA *(6) will follow an optimal bidding strategy $\hat{b}_i^*$ of,*

$$\hat{b}_i^* := p_\epsilon + v_i^p F_v(v_i^p) - \int_0^{v_i^p} F_v(z)dz > p_\epsilon, \tag{7}$$

*where $F_v(\cdot)$ denotes the cumulative density function of the random premium reward variable corresponding to the premium reward $v_i^p = V_i \lambda_i$.*

Theorem 2 applies to any distribution for $V_i$ and $\lambda_i$ on $[0, 1]$ and $[0, 1/2]$, i.e., $V_i \sim \mathcal{D}_V(0, 1)$ and $\lambda_i \sim \mathcal{D}_\lambda(0, 1/2)$, respectively. Determining specific optimal bids, utility, and model safety levels requires given distributions for $V_i$ and $\lambda_i$. Analysis of all-pay auctions (Amann & Leininger, 1996; Bhaskar, 2018; Tardos, 2017), as well as many other types of auctions, often assume a Uniform distribution for valuations. Therefore, our first analysis of SIRA, below in Corollary 1, presumes Uniform distributions for $V_i$ and $\lambda_i$.

**(Special Case 1) Uniform $V_i$ and $\lambda_i$: Optimal Agent Strategy.** Corollary 1 determines that a participating agent's optimal strategy to maximize its utility is to submit a model with safety levels larger than $\epsilon$ when their values $V_i$ and $\lambda_i$ come from a Uniform distribution.

**Corollary 1** (Uniform Nash Bidding Equilibrium). *Under Assumption 2, for agents having total value $V_i$ and scaling factor $\lambda_i$ both stemming from a Uniform distribution, with $v_i^d = (1 - \lambda_i)V_i$, and $v_i^p = \lambda_i V_i$, their optimal bid and utility participating in* SIRA *(6) are,*

$$b_i^* := \min\{\hat{b}_i^*, 1\}, \quad \hat{b}_i^* = \begin{cases} p_\epsilon + \frac{(v_i^p)^2 \ln(p_\epsilon)}{p_\epsilon - 1} & \text{if } 0 \leq v_i^p \leq \frac{p_\epsilon}{2}, \\ p_\epsilon + \frac{8(v_i^p)^2(\ln(2v_i^p) - 1/2) + p_\epsilon^2}{8(p_\epsilon - 1)} & \text{if } \frac{p_\epsilon}{2} \leq v_i^p \leq \frac{1}{2}, \end{cases} \tag{8}$$

$$u_i(b_i^*; \mathbf{b}_{-i}) = \begin{cases} \frac{2(v_i^p)^2 \ln(p_\epsilon)}{p_\epsilon - 1} + v_i^d - b_i^* & \text{if } 0 \leq v_i^p \leq \frac{p_\epsilon}{2}, \\ \frac{2(v_i^p)^2(\ln(2p_\epsilon) - 1) + p_\epsilon}{p_\epsilon - 1} + v_i^d - b_i^* & \text{if } \frac{p_\epsilon}{2} \leq v_i^p \leq \frac{1}{2}. \end{cases} \tag{9}$$

*Agents participating in* SIRA *submit models with the following safety,*

$$s_i^* := \begin{cases} M^{-1}(b_i^*) > \epsilon & \text{if } u_i(b_i^*; \mathbf{b}_{-i}) > 0, \\ 0 \text{ (no model submission)} & \text{else.} \end{cases} \tag{10}$$

**(Special Case 2) Beta $V_i$ and Uniform $\lambda_i$: Optimal Agent Strategy.** In many instances, a more realistic distribution for $V_i$ is a Beta distribution with $\alpha, \beta = 2$. This choice of distribution looks more Gaussian, with the bulk of the probability density centered in the middle, and as such is realistic

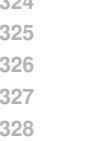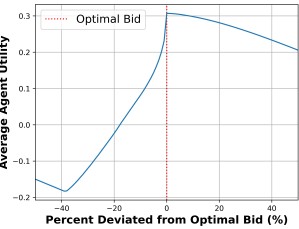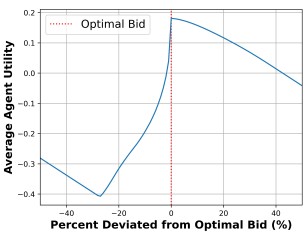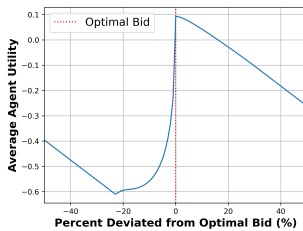

Figure 2: **Validation of Uniform Nash Bidding Equilibrium.** Agent utility is maximized when agents follow the theoretically optimal bidding function shown in Equation (8). Across varying safety prices, $p_\epsilon = 0.25$ (left), $0.5$ (middle), $0.75$ (right), agents attain less utility when they deviate from the optimal bid (red line) derived in Corollary 1.

when agent values do not congregate amongst one another and outliers (near 0 or 1) are rare. We analyze the performance of SIRA under this more realistic scenario in Corollary 2. Corollary 2 states that, under a Beta(2,2) distribution for $V_i$, agent $i$ maximizes its utility with an optimal bid $b_i^*$ larger than the price to attain $\epsilon$ safety, $b_i^* > p_\epsilon$, resulting in a model safer than the $\epsilon$ threshold. Furthermore, Corollaries 1 and 2 surpass the baseline optimal bid $b_i^* = p_\epsilon$ for Reserve Thresholding (Theorem 1).

**Corollary 2** (Beta Nash Bidding Equilibrium). *Under Assumption 2, let agents have total value $V_i$ and scaling factor $\lambda_i$ stem from Beta ($\alpha = \beta = 2$) and Uniform distributions respectively, with $v_i^d = (1 - \lambda_i)V_i$ and $v_i^p = \lambda_i V_i$. Denote the CDF of the Beta distribution on $[0, 1]$ as $F_\beta(x) = 3x^2 - 2x^3$. The optimal bid and utility for agents participating in SIRA (6) are,*

$$b_i^* := \min\{\hat{b}_i^*, 1\}, \quad \hat{b}_i^* = \begin{cases} p_\epsilon + \frac{3(v_i^p)^2(p_\epsilon^2 - 2p_\epsilon + 1)}{1 - F_\beta(p_\epsilon)} & if\ 0 \leq v_i^p \leq \frac{p_\epsilon}{2}, \\ p_\epsilon + \frac{8(v_i^p)^2\left(6(v_i^p)^2 - 8v_i^p + 3\right) + p_\epsilon^3(3p_\epsilon - 4)}{8(1 - F_\beta(p_\epsilon))} & if\ \frac{p_\epsilon}{2} \leq v_i^p \leq \frac{1}{2}, \end{cases} \quad (11)$$

$$u(b_i^*; \boldsymbol{b}_{-i}) = \begin{cases} v_i^d + \frac{6(v_i^p)^2(p_\epsilon^2 - 2p_\epsilon + 1)}{1 - F_\beta(p_\epsilon)} - b_i^* & for\ 0 \leq v_i^p \leq \frac{p_\epsilon}{2}, \\ v_i^d + \frac{v_i^p\left(8(v_i^p)^3 - 12(v_i^p)^2 + 6v_i^p + p_\epsilon^2(2p_\epsilon - 3)\right)}{1 - F_\beta(p_\epsilon)} - b_i^* & for\ \frac{p_\epsilon}{2} \leq v_i^p \leq 1/2. \end{cases} \quad (12)$$

*Agents participating in SIRA submit models with the following safety,*

$$s_i^* = \begin{cases} M^{-1}(b_i^*) > \epsilon & if\ u_i(b_i^*; \boldsymbol{b}_{-i}) > 0, \\ 0\ (no\ model\ submission) & else. \end{cases} \quad (13)$$

**Remark 2** (Improved Model Safety). *As shown in Corollaries 1 and 2, participating agents will submit models that are safer than the regulator's safety threshold, $s_i^* = M^{-1}(b_i^*) > \epsilon$.*

**Remark 3** (Improved Utility & Participation). *Through the introduction of regulator's premium compensation, agent utility is improved, in Equations (9) and (12), versus Reserve Thresholding in Equation (3). As a result, more agents break the zero-utility barrier of entry for participation, boosting both overall agent utility and participation rate.*

Due to space constraints, we place the proofs of Theorems 1 and 2 as well as Corollaries 1 and 2 within Appendix B. We note that since the premium compensation value $v_i^p$ is a product of two random variables, the PDF and CDF of $v_i^p$ becomes a piece-wise function (as shown within Appendix B). As a result, the optimal bidding and subsequent utility also becomes piece-wise in both Corollaries 1 and 2. We empirically verify the correctness of our computed PDF and CDFs within Appendix C.

## 6 EXPERIMENTS

Our Section 5 results demonstrate that this safety-incentivized regulatory auction, SIRA, creates incentives for any agents to submit safer models and to participate at higher rates than the baseline Reserve Thresholding mechanism in Section 4. Below, we validate these theoretical results empirically.

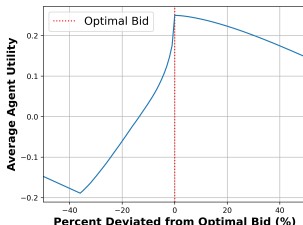 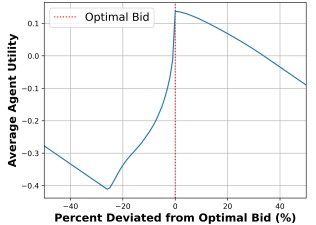 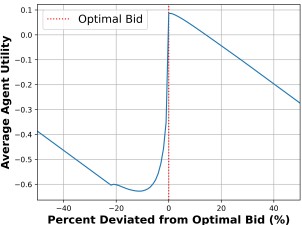

Figure 3: **Validation of Beta Nash Bidding Equilibrium.** Akin to the Uniform results, agent utility is maximized when agents follow the theoretically optimal bidding function shown in Equation (11). Across varying prices of safety values, $p_\epsilon = 0.25$ (left), 0.5 (middle), 0.75 (right), agents attain less utility when they deviate from the optimal bid (red line) derived in Corollary 2.

**Experimental Setup**. We simulate a regulatory setting with $n = 100,000$ agents. Each agent $i$ receives a random total value $V_i$ from either a Uniform (Corollary 1) or Beta(2,2) (Corollary 2) distribution. Each agent also receives a scaling factor $\lambda_i$ that splits the total value into deployment $v_i^d = (1 - \lambda_i)V_i$ and premium compensation $v_i^p = \lambda_i V_i$ values. Once the private values are provided, agents calculate their bid according to the optimal strategies in Theorem 1 and Corollaries 1 & 2.

**Lack of Baseline Regulatory Mechanisms**. To the best of our knowledge there are no other comparable mechanisms for safety regulation in AI. As a result, we compare against the Reserve Threshold mechanism that we propose in Section 4. While simple, the Reserve Threshold mechanism is a realistic baseline to compare against; there exist regulatory bodies, like the Environmental Protection Agency (EPA), that follow similar steps before clearing products (*e.g.,* the EPA authorizes permits for discharging pollutants into water sources once water quality criteria are met).

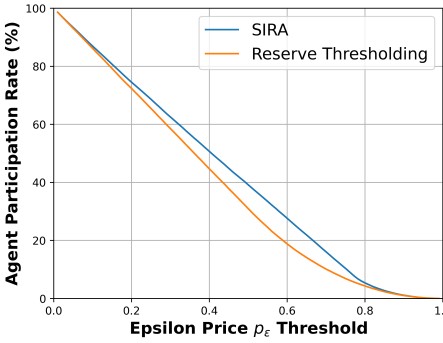 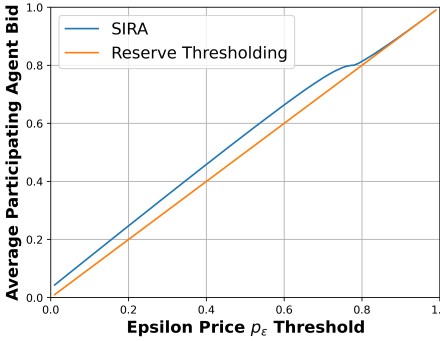

Figure 4: **Improved Safety and Participation with Uniform Values**. When total value stems from a Uniform distribution $V_i \sim U(0, 1)$, agents participate at a higher rate in SIRA than Reserve Thresholding (left) and spend more to train safer models (right).

**Verifiable Nash Bidding Equilibria**. Our first experimental goal is to validate that the theoretical bidding functions found in Corollaries 1 and 2 constitute Nash Equilibria. That is, agents receive worse utility if they deviate from this bidding strategy if all other agents abide by it. To test this, we compare the optimal bid for a single agent versus $100,000$ others. We vary the single agent's optimal bid on a range up to $\pm 50\%$. We note that comparisons only occur if the other agent's bid is at least $p_\epsilon$, in order to accurately reflect how the auction mechanism in Algorithm 1 functions.

In Figures 2 and 3, we plot the average utility over all $100,000$ comparisons. *We find that both our Uniform and Beta optimal bidding functions maximize agent utility and thus constitute Nash Equilibria.* Interestingly, utility decays much quicker when reducing the bid, since agents are **(i)** less likely to win the premium reward and **(ii)** at risk of losing the value from deployment if the bid does not reach $p_\epsilon$. At a certain point, utility increases linearly once the agent continuously fails to bid $p_\epsilon$. The linear improvement stems from the agent saving the cost of its bid, $-b_i$, shown in Equation (6).

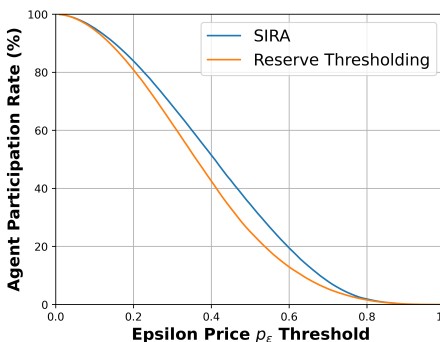 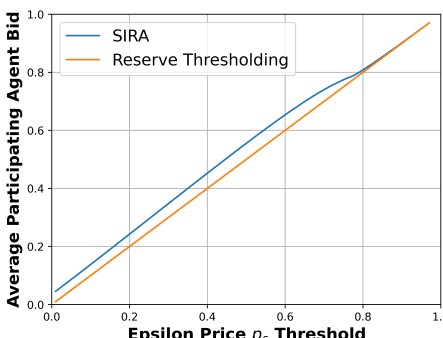

Figure 5: **Improved Safety and Participation with Beta Values**. When total value stems from a Beta distribution $V_i \sim \text{Beta}(\alpha = \beta = 2)$, agents participate at a higher rate in SIRA than Reserve Thresholding (left) and spend more to train safer models (right).

**Improved Agent Participation and Bid Size**. We find that for both Uniform and Beta(2,2) distributions, shown in Figures 4 and 5, our mechanism (SIRA) increases participation rates and average bids by upwards of 15% and 20% respectively. At the endpoints of possible price thresholds, $p_\epsilon = 0$ and 1, we find that both mechanisms perform similarly. The reason is that at a low safety threshold price $p_\epsilon \approx 0$, agents are highly likely to have a total value $V_i$ larger than a value close to zero. The inverse is true for $p_\epsilon \approx 1$, where it is unlikely that agents will have a total value $V_i$ larger than a value close to 1. Our mechanism shines when safety threshold prices are in the middle; the premium compensation offered by the regulator incentivizes agents to participate and bid more for the chance to win.

## 7 CONCLUSION AND FUTURE WORK

As AI models grow, the risks associated with their misuse become increasingly significant, particularly given their often opaque, black-box nature. Establishing robust algorithmic safeguards is crucial to protect users from unethical, unsafe, or illegally-deployed models. In this paper, we present a regulatory framework designed to ensure that only models deemed safe by a regulator can be deployed for public use. Our key contribution is the development of an auction-based regulatory mechanism that simultaneously (i) enforces safety standards and (ii) provably incentivizes agents to exceed minimum safety thresholds. This approach encourages broader participation and the development of safer models compared to baseline regulatory methods. Empirical results confirm that our mechanism increases agent participation by 15% and raises agent spending on safety by 20%, demonstrating its effectiveness to promote safer AI deployment.

**Future Work.** While this work addresses key challenges in regulating AI safety, several directions remain open for future exploration:

*(1) Model Evaluation:* Creating a realistic protocol for the regulator to evaluate submitted model safety levels is important to ensure agents do not skirt around safety requirements. While we leave this problem for future work, one possible solution is that agents can either provide the regulator API access to test its model or provide the model weights directly to the regulator. Truthfulness can be enforced via audits and the threat of legal action.

*(2) Extension to Heterogeneous Settings:* Extending our mechanism to heterogeneous scenarios, where evaluation data for agents and regulators differs, is a critical next step. Real-world data distributions often vary across contexts, and understanding how these variations affect both model safety and agent strategies will create a more robust regulatory mechanism. While explicit protocols or mathematical formulations are left as future work, we have a few ideas. One idea could be establishing a data-sharing framework between agents and the regulator, where each participating agent must contribute part of (or all of) its data to the regulator for evaluation. If data can be anonymized, then this would be a suitable solution. Another idea could be that the regulator collects data on its own, and can compare its distribution of data versus each participating agents' data distribution. If distributions greatly differ, then the regulator could collect more data or resort to the previous data-sharing method.

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

## A  NOTATION TABLE

Table 1: Notating and Defining all Variables Listed Within SIRA.

| Definition | Notation |
|---|---|
| Regulator | $R$ |
| Number of Agents | $n$ |
| Safety Threshold | $\epsilon$ |
| Safety-to-Cost Function | $M$ |
| Price of Attaining Safety | $p_\epsilon$ |
| Agent $i$ Bid | $b_i$ |
| All Other Agents Bids | $\boldsymbol{b}_{-i}$ |
| Agent $i$ Utility | $u_i$ |
| Agent $i$ Model Safety | $s_i$ |
| Total Value for Agent $i$ | $V_i$ |
| Total Value Distribution | $\mathcal{D}_V$ |
| Agent $i$ Scaling Factor | $\lambda_i$ |
| Scaling Factor Distribution | $\mathcal{D}_\lambda$ |
| Deployment Value for Agent $i$ | $v_i^d$ |
| Premium Compensation Value for Agent $i$ | $v_i^p$ |
| Probability Density Function for Premium Compensation | $f_v$ |
| Cumulative Distribution Function for Premium Compensation | $F_v$ |

## B  THEORETICAL PROOFS

Below, we provide the full proofs of our Theorems and Corollaries presented within our work.

### B.1  PROOF OF THEOREM 1

**Theorem 1** (Restated). *Under Assumption 2, agents participating in Reserve Thresholding (2) have an optimal bidding strategy and utility of,*

$$b_i^* = p_\epsilon, \quad u_i(b_i^*; \boldsymbol{b}_{-i}) = v_i^d - p_\epsilon,$$

*and submit models with the following safety level,*

$$s_i^* = \begin{cases} \epsilon & \text{if } u_i(b_i^*; \boldsymbol{b}_{-i}) > 0, \\ 0 \text{ (no model submission)} & \text{else.} \end{cases}$$

*Proof.* From agent $i$'s utility within Reserve Thresholding, Equation (2), it is clear that $u_i(0) = 0$. We proceed to break the proof up into cases where agents have (1) a deployment value equal to or less than the price of safety $p_\epsilon$ and (2) a deployment value larger than $p_\epsilon$.

**Case 1:** $v_i^d \le p_\epsilon$. From Equation (2), if $v_i^d \le p_\epsilon$ then an agent will never attain positive utility,

$$\max_{b_i \in (0,1]} v_i^d \cdot 1_{b_i \ge p_\epsilon} - b_i \le \max_{b_i \in (0,1]} p_\epsilon \cdot 1_{b_i \ge p_\epsilon} - b_i = \max_{b_i \in [p_\epsilon,1]} p_\epsilon - b_i = p_\epsilon - p_\epsilon = 0. \quad (14)$$

$$\arg\max_{b_i \in (0,1]} u_i(b_i) = p_\epsilon. \quad (15)$$

For an agent with deployment value at most equal to $p_\epsilon$, the upper bound on attainable utility when it participates, *i.e.,* $b_i \in (0,1]$, is zero (Equation (14)). This maximum utility is attained when bidding $b_i = p_\epsilon$ (Equation (15)). Thus, agents have nothing to gain by participating, as they already start at zero utility $u_i(0) = 0$. As a result, agents will not submit a model, $s_i^* = M(0) = 0$.

**Case 2:** $v_i^d > p_\epsilon$. Similar steps to Case 1 above,

$$\max_{b_i \in (0,1]} v_i^d \cdot 1_{b_i \geq p_\epsilon} - b_i > \max_{b_i \in (0,1]} p_\epsilon \cdot 1_{b_i \geq p_\epsilon} - b_i = \max_{b_i \in [p_\epsilon,1]} p_\epsilon - b_i = p_\epsilon - p_\epsilon = 0. \quad (16)$$

$$b_i^* = \arg\max_{b_i \in (0,1]} u_i(b_i) = p_\epsilon \longrightarrow u_i(b_i^*) = v_i^d - p_\epsilon > 0. \quad (17)$$

An agent with deployment value larger than $p_\epsilon$ will have a maximal utility that is non-negative when it participates (Equation (16)). Maximal utility is attained when bidding $b_i^* = p_\epsilon$ (Equation (17)). Furthermore, at this optimal bid, the corresponding safety level is $s_i^* = M^{-1}(p_\epsilon) = \epsilon$.

$$\square$$

### B.2 PROOF OF THEOREM 2

**Theorem 2** (Restated). *Agents participating in* SIRA (6) *will follow an optimal bidding strategy* $\hat{b}_i^*$ *of,*

$$\hat{b}_i^* := p_\epsilon + v_i^p F_v(v_i^p) - \int_0^{v_i^p} F_v(z)dz > p_\epsilon,$$

*where $F_v(\cdot)$ denotes the cumulative density function of the random variable corresponding to the premium reward $v_i^p = V_i\lambda_i$.*

*Proof.* Before beginning our proof, we note that each agent $i$ cannot alter its own valuation $v_i^p$ for winning the all-pay auction. Each valuation is private (unknown by other agents) and predetermined: total reward $V_i$ and partition factor $\lambda_i$ are randomly selected from a given distribution $\mathcal{D}$ on $[0, 1]$ and $[0, 1/2]$ respectively by "nature". We define the cumulative distribution function for the auction reward $v_i^p = V_i\lambda_i$ as $F_v(\cdot)$ and the probability distribution function as $f_v(\cdot)$.

From Equation (6), we find that an agent $i$ that does not participate (*i.e.,* $b_i = 0$) receives no utility,

$$u_i(0) = 0. \quad (18)$$

An agent receives negative utility if its bid does not reach the price of safety $p_\epsilon$,

$$\max_{b_i \in (0, p_\epsilon)} u_i(b_i) < 0. \quad (19)$$

Consequently, rational agents will either opt not to participate (notated as the set of agents $N$) or participate (notated as the set of agents $P$) and bid at least $p_\epsilon$. We define these groups as,

$$N = \{i \in [n] \mid \max_{b_i \in [0,1]} u_i(b_i) \leq 0\}, \quad (20)$$

$$P = \{i \in [n] \mid \max_{b_i \in [0,1]} u_i(b_i) > 0\}. \quad (21)$$

From here, we only focus on agents $i \in P$ which participate (*i.e.,* have utility to be gained by participating). As a result from Equations (18) and (19), Equation (21) transforms into,

$$P = \{i \in [n] \mid \max_{b_i \in [p_\epsilon,1]} u_i(b_i) > 0\}. \quad (22)$$

The result of (22) is that participating agents bid at least $p_\epsilon$. This is important, as every participating agent knows that all rival agents $j$ they will possibly be compared against have $b_j \in [p_\epsilon, 1]$. Agents can dictate how much they bid, and we design our auction to ensure that agents bid in proportion to their valuation.

Following previous literature (Amann & Leininger, 1996; Bhaskar, 2018; Tardos, 2017), we desire a *monotone increasing* bidding function $b(\cdot) : [0, 1/2] \to [p_\epsilon, 1]$ that each agent follows. We will prove that each agent $i$'s best strategy is to bid its own valuation $b(v_i^p)$ irrespective of other agent bids (Nash Equilibrium). Using a bidding function transforms agent utility,

$$u_i(b_i) = \left(v_i^d + v_i^p \cdot 1_{(\text{if } i \text{ wins auction})}\right) \cdot \underbrace{1_{(\text{if } b_i \geq p_\epsilon)}}_{\text{satisfied for agents } i \in P} - b_i,$$

$$= \mathbb{P}\left(b(b_i) > b(b_j)\right)v_i^p - b(b_i) + v_i^d, \quad b_j \sim \text{randomly sampled agent bid.} \quad (23)$$

Since $b(x)$ is monotone increasing up to 1, agents bidding $b = 1$ automatically win, the utility function above can be simplified as,

$$u_i(b_i) = v_i^p \mathbb{P}(b_i > b_j) - b(b_i) + v_i^d, \quad b_j \sim \text{randomly sampled agent bid,}$$

$$= v_i^p F_v(b_i) - b(b_i) + v_i^d. \tag{24}$$

Taking the derivative and setting it equal to zero yields,

$$\frac{d}{db_i} u_i(b_i) = v_i^p f_v(b_i) - b'(b_i) = 0. \tag{25}$$

As agents bid in proportion to their valuation, we solve the first-order equilibrium conditions at $b_i = v_i^p$,

$$b'(v_i^p) = v_i^p f_v(v_i^p). \tag{26}$$

Integrating by parts, and knowing $\epsilon$ is the minimum bid ($b(0) = p_\epsilon$), reveals our optimal bidding function,

$$b(v_i^p) - b(0) = \int_0^x v_i^p f_v(v_i^p) dv_i^p,$$

$$b(v_i^p) - p_\epsilon = v_i^p F_v(v_i^p) - \int_0^{v_i^p} F_v(z) dz,$$

$$\hat{b}_i^* = b(v_i^p) := p_\epsilon + v_i^p F_v(v_i^p) - \int_0^{v_i^p} F_v(z) dz. \tag{27}$$

$\square$

### B.3 Proof of Corollary 1

**Corollary 1** (Restated). *Under Assumption 2, for agents having total value $V_i$ and scaling factor $\lambda_i$ both stemming from a Uniform distribution, with $v_i^d = (1 - \lambda_i)V_i$, and $v_i^p = \lambda_i V_i$, their optimal bid and utility participating in* SIRA *(6) are,*

$$b_i^* := \min\{\hat{b}_i^*, 1\}, \quad \hat{b}_i^* = \begin{cases} p_\epsilon + \frac{(v_i^p)^2 \ln(p_\epsilon)}{p_\epsilon - 1} & \text{if } 0 \leq v_i^p \leq \frac{p_\epsilon}{2}, \\ p_\epsilon + \frac{8(v_i^p)^2(\ln(2v_i^p) - 1/2) + p_\epsilon^2}{8(p_\epsilon - 1)} & \text{if } \frac{p_\epsilon}{2} \leq v_i^p \leq \frac{1}{2}, \end{cases}$$

$$u_i(b_i^*; \boldsymbol{b}_{-i}) = \begin{cases} \frac{2(v_i^p)^2 \ln(p_\epsilon)}{p_\epsilon - 1} + v_i^d - b_i^* & \text{if } 0 \leq v_i^p \leq \frac{p_\epsilon}{2}, \\ \frac{2(v_i^p)^2(\ln(2p_\epsilon) - 1) + p_\epsilon}{p_\epsilon - 1} + v_i^d - b_i^* & \text{if } \frac{p_\epsilon}{2} \leq v_i^p \leq \frac{1}{2}. \end{cases}$$

*Agents participating in* SIRA *submit models with the following safety,*

$$s_i^* := \begin{cases} M^{-1}(b_i^*) > \epsilon & \text{if } u_i(b_i^*; \boldsymbol{b}_{-i}) > 0, \\ 0 \text{ (no model submission)} & \text{else.} \end{cases}$$

*Proof.* Let $v_i^p := V_i \lambda_i$, where $V_i \sim U[p_\epsilon, 1]$ and $\lambda_i \sim U[0, 1/2]$. The reason that $V_i$ is within the interval $[p_\epsilon, 1]$, is that all participating agents must have a value of at least $p_\epsilon$ or else they would not have rationale to bid. The smallest value of $V_i$ such that this is possible is $p_\epsilon$, so it is the lower bound on this interval. Our first goal is to find the PDF of $v_i^p$, $f_{v_i^p}(\cdot)$.

We begin solving for $f_{v_i^p}(\cdot)$ by using a change of variables. For the product of two random variables $v = x_1 \cdot x_2$, let $y_1 = x_1 \cdot x_2$ and $y_2 = x_2$. Thus, we find inversely that $x_2 = y_2$ and $x_1 = y_1/y_2$. Since $x_1$ and $x_2$ are independent and both uniform, we find that,

$$f_{y_1, y_2}(x_1, x_2) = \left(\frac{1}{1 - p_\epsilon}\right)\left(\frac{1}{1/2 - 0}\right) = \frac{2}{1 - p_\epsilon}. \tag{28}$$

When using the change of variables this becomes,

$$f_{y_1, y_2}(y_1, y_2) = f_{y_1, y_2}(x_1, x_2)|J| = \frac{2}{(1 - p_\epsilon)y_2}, \quad |J| = \left| \begin{pmatrix} 1/y_2 & -y_1/y_2^2 \\ 0 & 1 \end{pmatrix} \right| = 1/y_2 \tag{29}$$

Marginalizing out $y_2$ (a non-negative value) yields,

$$f_{y_1}(y_1) = \int_0^\infty \frac{2}{(1-p_\epsilon)y_2} dy_2. \tag{30}$$

The bounds of integration depend upon the value of $y_1$. The change of variable to the $(y_1, y_2)$ space, where $0 \leq y_1, y_2 \leq 1/2$, results in a new region of possible variable values. This region is a triangle bounded by the three vertices: $(0,0)$, $(p_\epsilon/2, 1/2)$, and $(1/2, 1/2)$. Thus, the bounds of marginalization depend upon the value of $y_1$. For $0 \leq y_1 \leq p_\epsilon/2$ we have,

$$f_{y_1}(y_1) = \int_{y_1}^{y_1/p_\epsilon} \frac{2}{(1-p_\epsilon)y_2} dy_2 = \frac{2}{(1-p_\epsilon)}[\ln(y_2)|_{y_1}^{y_1/p_\epsilon}] = \frac{2\ln(p_\epsilon)}{(p_\epsilon - 1)}. \tag{31}$$

For $p_\epsilon \leq y_1 \leq 1/2$ we have,

$$f_{y_1}(y_1) = \int_{y_1}^{1/2} \frac{2}{(1-p_\epsilon)y_2} dy_2 = \frac{2}{(1-p_\epsilon)}[\ln(y_2)|_{y_1}^{1/2}] = \frac{2\ln(2y_1)}{(p_\epsilon - 1)}. \tag{32}$$

Thus, as a piecewise function the PDF is formally,

$$f_{y_1}(y_1) = \begin{cases} \frac{2\ln(p_\epsilon)}{(p_\epsilon-1)} & \text{for } 0 \leq y_1 \leq \frac{p_\epsilon}{2}, \\ \frac{2\ln(2y_1)}{(p_\epsilon-1)} & \text{for } \frac{p_\epsilon}{2} \leq y_1 \leq 1/2. \end{cases} \tag{33}$$

Now, the CDF is determined through integration,

$$F_{y_1}(y_1) = \int_0^{y_1} f_{y_1}(y_1) dy_1 = \begin{cases} \frac{2y_1 \ln(p_\epsilon)}{(p_\epsilon-1)} & \text{for } 0 \leq y_1 \leq \frac{p_\epsilon}{2}, \\ \frac{2y_1(\ln(2y_1)-1)+p_\epsilon}{(p_\epsilon-1)} & \text{for } \frac{p_\epsilon}{2} \leq y_1 \leq 1/2. \end{cases} \tag{34}$$

We can integrate the CDF to get,

$$\int_0^{y_1} F_{y_1}(y_1) = \begin{cases} \frac{y_1^2 \ln(p_\epsilon)}{(p_\epsilon-1)} & \text{for } 0 \leq y_1 \leq \frac{p_\epsilon}{2}, \\ \frac{4y_1^2(2\ln(2y_1)-3)+8y_1 p_\epsilon - p_\epsilon^2}{8(p_\epsilon-1)} & \text{for } \frac{p_\epsilon}{2} \leq y_1 \leq 1/2. \end{cases} \tag{35}$$

Plugging all of this back into Equation (7) yields,

$$\hat{b}_i^* = \begin{cases} p_\epsilon + v_i^p \frac{2v_i^p \ln(p_\epsilon)}{p_\epsilon - 1} - \frac{(v_i^p)^2 \ln(p_\epsilon)}{p_\epsilon - 1}, \\ p_\epsilon + v_i^p \frac{2v_i^p(\ln(2v_i^p)-1)+p_\epsilon}{(p_\epsilon-1)} - \frac{4(v_i^p)^2(2\ln(2v_i^p)-3)+8v_i^p p_\epsilon - p_\epsilon^2}{8(p_\epsilon-1)}, \end{cases}$$

$$= \begin{cases} p_\epsilon + \frac{(v_i^p)^2 \ln(p_\epsilon)}{p_\epsilon - 1} & \text{if } 0 \leq v_i^p \leq \frac{p_\epsilon}{2}, \\ p_\epsilon + \frac{8(v_i^p)^2(\ln(2v_i^p)-1/2)+p_\epsilon^2}{8(p_\epsilon-1)} & \text{if } \frac{p_\epsilon}{2} \leq v_i^p \leq \frac{1}{2}. \end{cases} \tag{36}$$

Since $b_i$ cannot be larger than 1, we cap the bidding function at one via,

$$b_i^* := \min\{\hat{b}_i^*, 1\}. \tag{37}$$

The utility gained by agent $i$ for using such a bidding function is,

$$u(b_i^*) = \begin{cases} v_i^d - b_i^* + \frac{2(v_i^p)^2 \ln(p_\epsilon)}{p_\epsilon - 1} & \text{for } 0 \leq v_i^p \leq \frac{p_\epsilon}{2}, \\ v_i^d - b_i^* + \frac{2(v_i^p)^2(\ln(2v_i^p)-1)+p_\epsilon}{(p_\epsilon-1)} & \text{for } \frac{p_\epsilon}{2} \leq v_i^p \leq 1/2. \end{cases} \tag{38}$$

When this utility is larger than 0, the agent will participate otherwise the agent will not submit a model to the regulator. Finally, we can find the optimal safety level by using Assumption 2,

$$s_i^* := M^{-1}(b_i^*). \tag{39}$$

$\square$

## B.4  PROOF OF COROLLARY 2

**Corollary 2** (Restated). *Under Assumption 2, let agents have total value $V_i$ and scaling factor $\lambda_i$ stem from Beta ($\alpha = \beta = 2$) and Uniform distributions respectively, with $v_i^d = (1 - \lambda_i)V_i$ and $v_i^p = \lambda_i V_i$. Denote the CDF of the Beta distribution on $[0, 1]$ as $F_\beta(x) = 3x^2 - 2x^3$. The optimal bid and utility for agents participating in* SIRA *(6) are,*

$$
b_i^* := \min\{\hat{b}_i^*, 1\}, \quad \hat{b}_i^* = \begin{cases} p_\epsilon + \frac{3(v_i^p)^2(p_\epsilon^2 - 2p_\epsilon + 1)}{1 - F_\beta(p_\epsilon)} & \text{if } 0 \leq v_i^p \leq \frac{p_\epsilon}{2}, \\ p_\epsilon + \frac{8(v_i^p)^2\left(6(v_i^p)^2 - 8v_i^p + 3\right) + p_\epsilon^3(3p_\epsilon - 4)}{8(1 - F_\beta(p_\epsilon))} & \text{if } \frac{p_\epsilon}{2} \leq v_i^p \leq \frac{1}{2}, \end{cases}
$$

$$
u(b_i^*; \boldsymbol{b}_{-i}) = \begin{cases} v_i^d + \frac{6(v_i^p)^2(p_\epsilon^2 - 2p_\epsilon + 1)}{1 - F_\beta(p_\epsilon)} - b_i^* & \text{for } 0 \leq v_i^p \leq \frac{p_\epsilon}{2}, \\ v_i^d + \frac{v_i^p\left(8(v_i^p)^3 - 12(v_i^p)^2 + 6v_i^p + p_\epsilon^2(2p_\epsilon - 3)\right)}{1 - F_\beta(p_\epsilon)} - b_i^* & \text{for } \frac{p_\epsilon}{2} \leq v_i^p \leq 1/2. \end{cases}
$$

*Agents participating in* SIRA *submit models with the following safety,*

$$
s_i^* = \begin{cases} M^{-1}(b_i^*) > \epsilon & \text{if } u_i(b_i^*; \boldsymbol{b}_{-i}) > 0, \\ 0 \text{ (no model submission)} & \text{else.} \end{cases}
$$

*Proof.* Similar to Corollary 1, we begin solving for $f_{v_i^p}(\cdot)$ using a change of variables. For the product of two random variables $v = x_1 \cdot x_2$, let $y_1 = x_1 \cdot x_2$ and $y_2 = x_2$. Inversely, $x_2 = y_2$ and $x_1 = y_1/y_2$. While $x_1$ and $x_2$ are independent, $x_1$ comes from a Beta distribution and $x_2$ from a Uniform one. The PDF and CDF of a Beta distribution, with $\alpha = \beta = 2$, on $[0, 1]$ are defined as,

$$
f_\beta(x) := 6x(1 - x), \tag{40}
$$

$$
F_\beta(x) := 3x^2 - 2x^3. \tag{41}
$$

Now, the PDF over $y_1, y_2$ is defined as,

$$
f_{y_1, y_2}(x_1, x_2) = \left(\frac{6x_1(1 - x_1)}{1 - F_\beta(p_\epsilon)}\right)\left(\frac{1}{1/2 - 0}\right) = \frac{12x_1(1 - x_1)}{1 - F_\beta(p_\epsilon)}. \tag{42}
$$

When using the change of variables this becomes,

$$
f_{y_1, y_2}(y_1, y_2) = f_{y_1, y_2}(x_1, x_2)|J| = \frac{12y_1(1 - \frac{y_1}{y_2})}{(1 - F_\beta(p_\epsilon))y_2^2}, \quad |J| = \left|\begin{pmatrix} 1/y_2 & -y_1/y_2^2 \\ 0 & 1 \end{pmatrix}\right| = 1/y_2 \tag{43}
$$

Marginalizing out $y_2$ (a non-negative value) yields,

$$
f_{y_1}(y_1) = \frac{12y_1}{1 - F_\beta(p_\epsilon)} \int_0^\infty \frac{1}{y_2^2} - \frac{y_1}{y_2^3} dy_2. \tag{44}
$$

The bounds of integration depend upon the value of $y_1$. The change of variable to the $(y_1, y_2)$ space, where $0 \leq y_1, y_2 \leq 1/2$, results in a new region of possible variable values. This region is a triangle bounded by the three vertices: $(0, 0)$, $(p_\epsilon/2, 1/2)$, and $(1/2, 1/2)$. Thus, the bounds of marginalization depend upon the value of $y_1$. For $0 \leq y_1 \leq p_\epsilon/2$ we have,

$$
f_{y_1}(y_1) = \frac{12y_1}{1 - F_\beta(p_\epsilon)} \int_{y_1}^{y_1/p_\epsilon} \frac{1}{y_2^2} - \frac{y_1}{y_2^3} dy_2 = \frac{12y_1}{1 - F_\beta(p_\epsilon)}\left[-\frac{1}{y_2} + \frac{y_1}{2y_2^2}\Big|_{y_1}^{y_1/p_\epsilon}\right]
$$

$$
= \frac{12y_1}{1 - F_\beta(p_\epsilon)}\left[-\frac{p_\epsilon}{y_1} + \frac{p_\epsilon^2}{2y_1} + \frac{1}{y_1} - \frac{1}{2y_1}\right] = \frac{6(p_\epsilon^2 - 2p_\epsilon + 1)}{1 - F_\beta(p_\epsilon)}. \tag{45}
$$

For $p_\epsilon \leq y_1 \leq 1/2$ we have,

$$
f_{y_1}(y_1) = \frac{12y_1}{1 - F_\beta(p_\epsilon)} \int_{y_1}^{1/2} \frac{1}{y_2^2} - \frac{y_1}{y_2^3} dy_2 = \frac{12y_1}{1 - F_\beta(p_\epsilon)}\left[-\frac{1}{y_2} + \frac{y_1}{2y_2^2}\Big|_{y_1}^{1/2}\right]
$$

$$
= \frac{12y_1}{1 - F_\beta(p_\epsilon)}\left[-2 + 2y_1 + \frac{1}{y_1} - \frac{1}{2y_1}\right] = \frac{6(4y_1^2 - 4y_1 + 1)}{1 - F_\beta(p_\epsilon)}. \tag{46}
$$

Thus, as a piecewise function the PDF is formally,

$$f_{y_1}(y_1) = \begin{cases} \frac{6(p_\epsilon^2 - 2p_\epsilon + 1)}{1 - F_\beta(p_\epsilon)} & \text{for } 0 \le y_1 \le \frac{p_\epsilon}{2}, \\ \frac{6(4y_1^2 - 4y_1 + 1)}{1 - F_\beta(p_\epsilon)} & \text{for } \frac{p_\epsilon}{2} \le y_1 \le 1/2. \end{cases} \tag{47}$$

Now, the CDF is determined through integration,

$$F_{y_1}(y_1) = \int_0^{y_1} f_{y_1}(y_1) dy_1 = \begin{cases} \frac{6y_1(p_\epsilon^2 - 2p_\epsilon + 1)}{1 - F_\beta(p_\epsilon)} & \text{for } 0 \le y_1 \le \frac{p_\epsilon}{2}, \\ \frac{2y_1(4y_1^2 - 6y_1 + 3) + p_\epsilon^2(2p_\epsilon - 3)}{1 - F_\beta(p_\epsilon)} & \text{for } \frac{p_\epsilon}{2} \le y_1 \le 1/2. \end{cases} \tag{48}$$

We can integrate the CDF to get,

$$\int_0^{y_1} F_{y_1}(y_1) = \begin{cases} \frac{3y_1^2(p_\epsilon^2 - 2p_\epsilon + 1)}{1 - F_\beta(p_\epsilon)} & \text{for } 0 \le y_1 \le \frac{p_\epsilon}{2}, \\ \frac{8y_1\left(2y_1^3 - 4y_1^2 + 3y_1 + p_\epsilon^2(2p_\epsilon - 3)\right) + p_\epsilon^3(4 - 3p_\epsilon)}{8(1 - F_\beta(p_\epsilon))} & \text{for } \frac{p_\epsilon}{2} \le y_1 \le 1/2. \end{cases} \tag{49}$$

Plugging all of this back into Equation (7) yields,

$$\begin{aligned} \hat{b}_i^* &= \begin{cases} p_\epsilon + v_i^p \frac{6v_i^p(p_\epsilon^2 - 2p_\epsilon + 1)}{1 - F_\beta(p_\epsilon)} - \frac{3(v_i^p)^2(p_\epsilon^2 - 2p_\epsilon + 1)}{1 - F_\beta(p_\epsilon)}, \\ p_\epsilon + v_i^p \frac{2v_i^p(4(v_i^p)^2 - 6v_i^p + 3) + p_\epsilon^2(2p_\epsilon - 3)}{1 - F_\beta(p_\epsilon)} - \frac{8v_i^p\left(2(v_i^p)^3 - 4(v_i^p)^2 + 3v_i^p + p_\epsilon^2(2p_\epsilon - 3)\right) + p_\epsilon^3(4 - 3p_\epsilon)}{8(1 - F_\beta(p_\epsilon))}, \end{cases} \\ &= \begin{cases} p_\epsilon + \frac{3(v_i^p)^2(p_\epsilon^2 - 2p_\epsilon + 1)}{1 - F_\beta(p_\epsilon)} & \text{if } 0 \le v_i^p \le \frac{p_\epsilon}{2}, \\ p_\epsilon + \frac{8(v_i^p)^2\left(6(v_i^p)^2 - 8v_i^p + 3\right) + p_\epsilon^3(3p_\epsilon - 4)}{8(1 - F_\beta(p_\epsilon))} & \text{if } \frac{p_\epsilon}{2} \le v_i^p \le \frac{1}{2}. \end{cases} \end{aligned} \tag{50}$$

Since $b_i$ cannot be larger than 1, we cap the bidding function at one via,

$$b_i^* := \min\{\hat{b}_i^*, 1\}. \tag{51}$$

The utility gained by agent $i$ for using such a bidding function is,

$$u(b_i^*) = \begin{cases} v_i^d - b_i^* + \frac{6(v_i^p)^2(p_\epsilon^2 - 2p_\epsilon + 1)}{1 - F_\beta(p_\epsilon)} & \text{for } 0 \le v_i^p \le \frac{p_\epsilon}{2}, \\ v_i^d - b_i^* + \frac{v_i^p\left(8(v_i^p)^3 - 12(v_i^p)^2 + 6v_i^p + p_\epsilon^2(2p_\epsilon - 3)\right)}{1 - F_\beta(p_\epsilon)} & \text{for } \frac{p_\epsilon}{2} \le v_i^p \le 1/2. \end{cases} \tag{52}$$

When this utility is larger than 0, the agent will participate otherwise the agent will not submit a model to the regulator. Finally, we can find the optimal safety level by using Assumption 2,

$$s_i^* := M^{-1}(b_i^*). \tag{53}$$

$\square$

## C ADDITIONAL EXPERIMENTS

Within this section, we verify empirically that our computed PDF and CDFs in Corollaries 1 and 2 are correct. To accomplish this, we randomly sample and compute the product of $V_i$ and $\lambda_i$ fifty million times. We then plot the PDF and CDF of the resultant products and compare it with our theoretical PDF and CDF. The theoretical PDF and CDF for Corollary 1 are defined in Equations (33) and (34), while those for Corollary 2 are found in Equations (47) and (48). The results of these simulations, which validate our computed PDFs and CDFs, are shown in Figures 6 and 7. To ensure correctness, we perform testing on different values of $p_\epsilon$. As expected, our theory lines up exactly with our empirical simulations for both Corollaries as well as across varying $p_\epsilon$.

### C.1 SAFETY-COST ABLATION STUDY

In this section, we conduct an ablation study to demonstrate that in realistic settings, safety is mapped to cost in a monotonically increasing way (as detailed in Assumption 2). While there are many

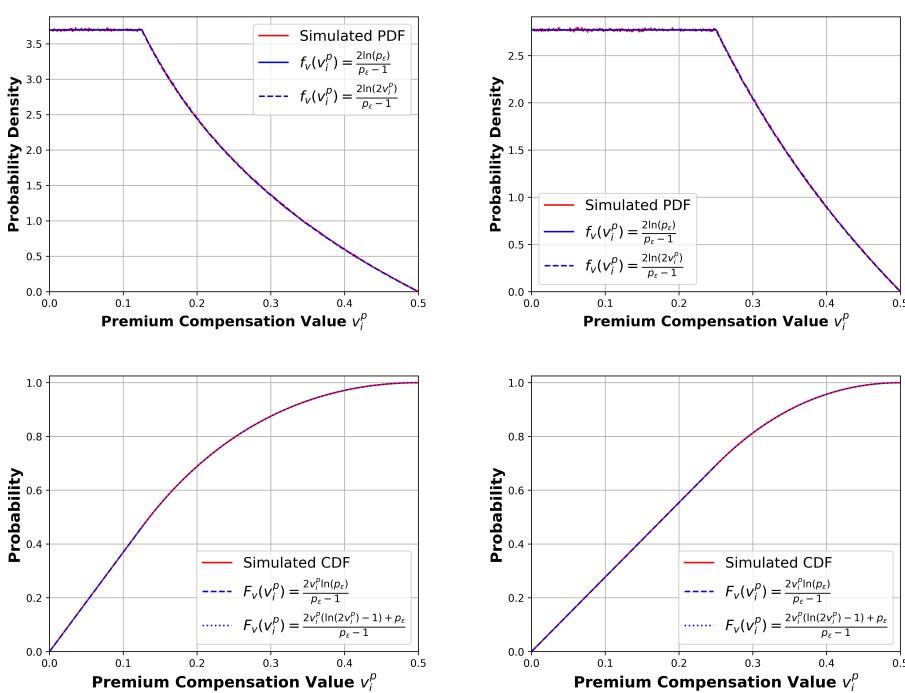

Figure 6: Numerical validation of our derivations for $f_v(v_i^p)$ and $F_v(v_i^p)$, where $v_i^p := V_i\lambda_i$, for $V_i$ and $\lambda_i$ coming from Uniform distributions (Corollary 1). The price of attaining $\epsilon$ is set as $p_\epsilon = 1/4$ (top row) and $p_\epsilon = 1/2$ (bottom row).

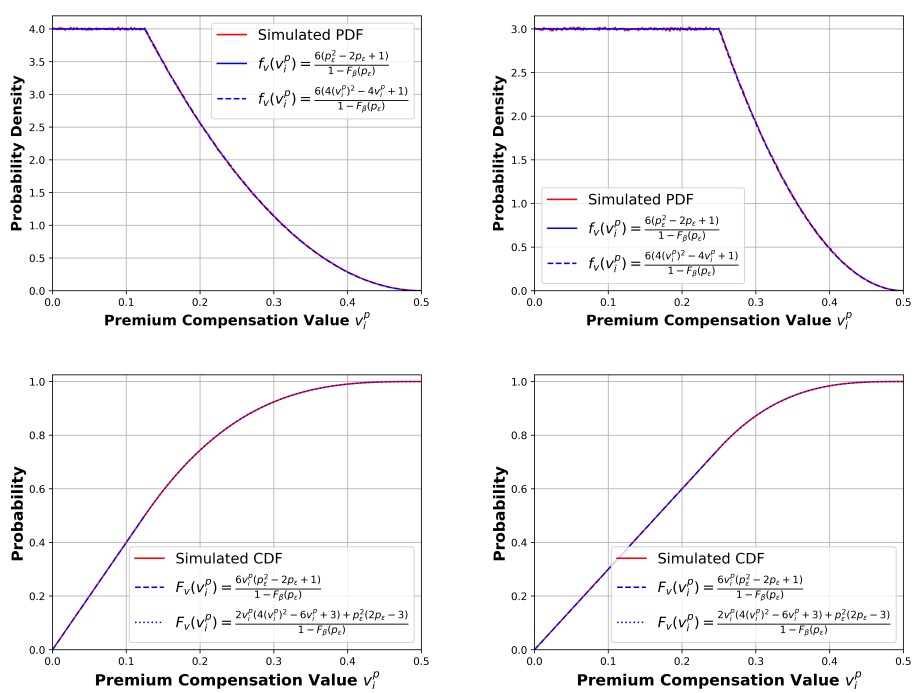

Figure 7: Numerical validation of our derivations for $f_v(v_i^P)$ and $F_v(v_i^P)$, where $v_i^p := V_i\lambda_i$, for $V_i$ coming from a Beta distribution and $\lambda_i$ from a Uniform distributions (Corollary 2). The price of attaining $\epsilon$ is set as $p_\epsilon = 1/4$ (top row) and $p_\epsilon = 1/2$ (bottom row).

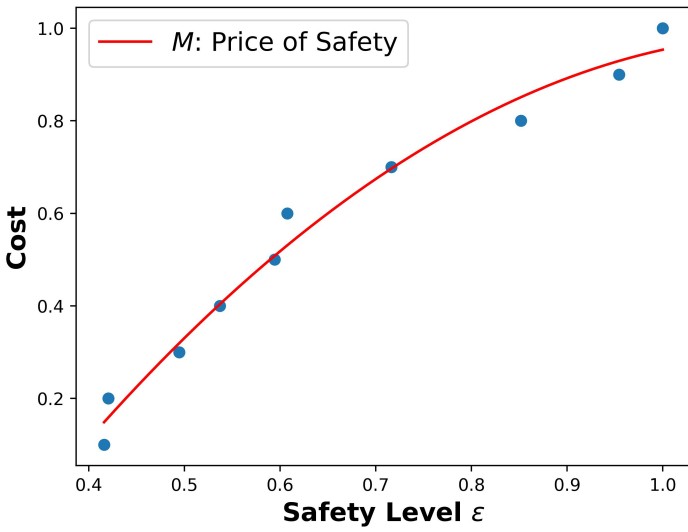

Figure 8: Strictly monotonic relationship between safety and cost. As the percentage of minority class data increases (greater cost), the equalized odds metric improves (greater safety) on Fairface.

factors to consider when gauging safe AI deployment, we analyze model fairness, via equalized odds, for image classification in this study. Equalized odds measures if different groups have similar true positive rates and false positive rates (lower is better). We train VGG-16 models on the Fairface dataset (Karkkainen & Joo, 2021) for 50 epochs (repeated ten times with different random seeds), and consider a gender classification task with race as the sensitive attribute. Models with the largest validation classification accuracy during training are selected for testing.

Many types of costs exist for training safer models, such as extensive architecture and hyper-parameter search. In this study, we consider the cost of an agent acquiring more minority class data. Acquiring more minority class data leads to a larger and more balanced dataset. We simulate various mixtures of training data, starting from a 95:5 skew and scaling up to fully balanced training data with respect to the sensitive attribute. In our study, we gauge equalized odds performance on well-balanced test data for the models trained on various mixtures of data. The results of this ablation study are shown in Table 2 and Figure 8.

Table 2: Equalized odds performance as minority class data increases.

| Minority Class % | Mean Equalized Odds Score |
|---|---|
| 5% | 22.55 |
| 10% | 22.31 |
| 15% | 18.97 |
| 20% | 17.46 |
| 25% | 15.78 |
| 30% | 15.44 |
| 35% | 13.09 |
| 40% | 11.01 |
| 45% | 9.83 |
| 50% | 9.38 |

As expected, in Table 2, the equalized odds score decreases (the model becomes safer) when collecting more minority class data (increased cost). To adjust equalized odds to fit into the setting where $\epsilon \in (0, 1)$, we inverted and normalized the original equalized odds score. In Figure 8, one can see that safety level is indeed monotonically increasing with respect to the cost.

# D    REPEATING SIRA AUCTIONS

The current auction structure (Algorithm 1) expects agents to submit a single model trained solely for the upcoming auction. There is no expectation that the model will be reused for a future auction, or indication that the model has been submitted to a previous auction. Looking towards the future, we would like to design SIRA to fit a repeatable auction structure, in which approved or rejected models may be resubmitted in subsequent auctions.

**Repeated Agent Utility**. Previously, in Algorithm 1, agents start the regulatory process with zero cost and value (*i.e.,* they are building their models from scratch). In repeating SIRA auctions, agent cost and value are accumulated across all previous auction submissions. For example, if an agent trains its already-accepted model further to attain a higher safety level $s_i$, its total accumulated training cost is $M(s_i)$. This agent's total value becomes the value its model gained from previous auction submissions plus any value gained from the current auction.

By allowing repeated SIRA auctions, an agent is able to repeatedly submit its model for regulatory review. We note that repeated submissions decrease the value of model deployment; once an agent earns the reward for deploying their model, subsequent deployments of the same model with improved safety levels can be realistically expected to earn less value than the initial deployment. We characterize this loss in value for repeated submissions with an indicator function in the utility function that only allows deployment value to be obtained once, on initial acceptance of a model. While we allow agents to win premium rewards across multiple auctions, we note that a regulator can curb this by either limiting the number of auction submissions per agent or the number of auctions held per year. We now define the repeated SIRA auction utility of agent $i$, who has participated in $a - 1$ previous auctions, as:

$$u_{i,a}(b_i) = \left( \sum_{n=1}^{a} \nu_i^n \right) - b_i, \tag{54}$$

where $\nu_i^n$, the value gained at the $n^{th}$ auction model $i$ was submitted to, is formulated as:

$$\nu_i^n = \begin{cases} v_i^{d,n} \cdot 1_{(\text{if } \nu_i^{n-1} = 0)} & \text{if } b_j^n \geq p_\epsilon^n \text{ and } b_i^n < b_j^n \text{ randomly sampled bid } b_j^n, \\ v_i^{d,n} \cdot 1_{(\text{if } \nu_i^{n-1} = 0)} + v_i^{p,n} & \text{if } b_i^n \geq p_\epsilon^n \text{ and } b_i^n > b_j^n \text{ randomly sampled bid } b_j^n, \\ 0 & \text{if } n \leq 0. \end{cases} \tag{55}$$

The repeated SIRA auction setup creates a unique property for models in training. If an agent intends to obtain a high safety level, but an auction takes place mid-training, the agent is actually incentivized to submit their model early if they have a chance at winning the premium reward. Though the model may have a lower likelihood of earning the reward, there is no consequence for models failing to attain the premium reward. Gaining value is strictly beneficial to agents, and accumulated value helps offset the costs of training a model. This property only exists for the premium reward; the deployment reward can only be obtained once, thus there is no incentive to submit early to earn it.

**Repeated Optimal Bidding Function**. Using the same assumptions for single-auction SIRA, namely Assumptions 1 and 2 along with private values, we can derive the bidding function for a rational agent under a repeated SIRA auction setting. We follow an equivalent setup to Lemma 1 with regards to the valuation of rewards, giving us the cumulative distribution function for $v_i^p = V_i \lambda_i$ as $F_v(\cdot)$ and the probability distribution function as $f_v(\cdot)$.

From our definition of utility $u_{i,a}(b_i)$, we find that an agent $i$ that does not participate (*i.e.,* submitting $b_i = 0$) receives utility equal to $\nu_i^a$. However, since $b_i = 0$ will never be larger than $p_\epsilon$ (by definition), it must be true that $\nu_i^a = 0$ as well, since the model will never meet the required safety threshold. Therefore, a non-participating agent will always receive non-negative utility.

$$u_{i,a}(0) = 0. \tag{56}$$

Following closely to the proof of Theorem 2 in Appendix B, we find that participating agents $i \in P$ (with $P$ defined in the previous proof) will now have a utility of,

$$u_{i,a}(b_i) = \nu_i^a + v_i^d \cdot 1_{(\nu_i^a = 0)} + v_i^p \mathbb{P}(b_i > b_j) - b(b_i), \quad b_j \sim \text{randomly sampled agent bid,}$$

$$= \nu_i^a + v_i^d \cdot 1_{(\nu_i^a = 0)} + v_i^p F_v(b_i) - b(b_i). \tag{57}$$

Taking the derivative and setting it equal to zero yields,

$$\frac{d}{db_i} u_{i,a}(b_i) = v_i^p f_v(b_i) - b'(b_i) = 0. \tag{58}$$

As agents bid in proportion to their valuation, we solve the first-order conditions at $b_i = v_i^p$,

$$b'(v_i^p) = v_i^p f_v(v_i^p). \tag{59}$$

Note, at this point in the proof the bidding function calculation is now equivalent to the calculations found in Lemma 1. We can thus follow the same steps to reveal our optimal bidding function,

$$b(v_i^p) := p_\epsilon + v_i^p F_v(v_i^p) - \int_0^{v_i^p} F_v(z)dz, \tag{60}$$

which is equivalent to the optimal bidding function derived in Lemma 1.

As the optimal bidding function is equivalent, calculations for the Nash Bidding Equilibrium are also equivalent to those found in Corollary 1 and Corollary 2. The optimal bid and utility participating in SIRA (6) under the assumptions of Corollary 1 will thus be,

$$b_i^* := \min\{\hat{b}_i^*, 1\}, \quad \hat{b}_i^* = \begin{cases} p_\epsilon + \frac{(v_i^p)^2 \ln(p_\epsilon)}{p_\epsilon - 1} & \text{if } 0 \le v_i^p \le \frac{p_\epsilon}{2}, \\ p_\epsilon + \frac{8(v_i^p)^2(\ln(2v_i^p) - 1/2) + p_\epsilon^2}{8(p_\epsilon - 1)} & \text{if } \frac{p_\epsilon}{2} \le v_i^p \le \frac{1}{2}, \end{cases}$$

$$u_{i,a}(b_i^*; \boldsymbol{b}_{-i}) = \begin{cases} \nu_i^a + v_i^d \cdot 1_{(\nu_i^a = 0)} + \frac{2(v_i^p)^2 \ln(p_\epsilon)}{p_\epsilon - 1} - b_i^* & \text{if } 0 \le v_i^p \le \frac{p_\epsilon}{2}, \\ \nu_i^a + v_i^d \cdot 1_{(\nu_i^a = 0)} + \frac{2(v_i^p)^2(\ln(2p_\epsilon) - 1) + p_\epsilon}{p_\epsilon - 1} - b_i^* & \text{if } \frac{p_\epsilon}{2} \le v_i^p \le \frac{1}{2}. \end{cases}$$

Agents participating in SIRA under Corollary 1 submit models with the following safety,

$$s_i^* := \begin{cases} M^{-1}(b_i^*) > \epsilon & \text{if } u_i(b_i^*; \boldsymbol{b}_{-i}) > 0, \\ 0 \text{ (no model submission)} & \text{else.} \end{cases}$$

The optimal bid and utility participating in SIRA (6) under the assumptions of Corollary 2 will be,

$$b_i^* := \min\{\hat{b}_i^*, 1\}, \quad \hat{b}_i^* = \begin{cases} p_\epsilon + \frac{3(v_i^p)^2(p_\epsilon^2 - 2p_\epsilon + 1)}{1 - F_\beta(p_\epsilon)} & \text{if } 0 \le v_i^p \le \frac{p_\epsilon}{2}, \\ p_\epsilon + \frac{8(v_i^p)^2\left(6(v_i^p)^2 - 8v_i^p + 3\right) + p_\epsilon^3(3p_\epsilon - 4)}{8(1 - F_\beta(p_\epsilon))} & \text{if } \frac{p_\epsilon}{2} \le v_i^p \le \frac{1}{2}, \end{cases}$$

$$u_{i,a}(b_i^*; \boldsymbol{b}_{-i}) = \begin{cases} \nu_i^a + v_i^d \cdot 1_{(\nu_i^a = 0)} + \frac{6(v_i^p)^2(p_\epsilon^2 - 2p_\epsilon + 1)}{1 - F_\beta(p_\epsilon)} - b_i^* & \text{for } 0 \le v_i^p \le \frac{p_\epsilon}{2}, \\ \nu_i^a + v_i^d \cdot 1_{(\nu_i^a = 0)} + \frac{v_i^p\left(8(v_i^p)^3 - 12(v_i^p)^2 + 6v_i^p + p_\epsilon^2(2p_\epsilon - 3)\right)}{1 - F_\beta(p_\epsilon)} - b_i^* & \text{for } \frac{p_\epsilon}{2} \le v_i^p \le 1/2. \end{cases}$$

Agents participating in SIRA under Corollay 2 submit models with the following safety,

$$s_i^* = \begin{cases} M^{-1}(b_i^*) > \epsilon & \text{if } u_i(b_i^*; \boldsymbol{b}_{-i}) > 0, \\ 0 \text{ (no model submission)} & \text{else.} \end{cases}$$

