# OpenReview forum: "Auction-Based Regulation for Artificial Intelligence"
_ICLR.cc/2025/Conference — Submitted to ICLR 2025_

### Official Review · Reviewer_F4Wy · 2024-10-22

**Soundness:** 2
**Presentation:** 2
**Contribution:** 2
**Rating:** 5
**Confidence:** 4

**Summary:**

The authors provides a formulation of the AI regulatory process as an all-pay auction, and design an auction-based regulatory mechanism that produces Nash Equilibria that induces safety considerations.

**Strengths:**

- A novel and important question, and strong motivation
- Sound theoretical analysis
- Genrally well-written

**Weaknesses:**

The authors' formulation of the regulatory process and safety components appears to be somewhat simplified and may diverge from current AI developments in a few key ways:
- (Minor) The authors assume a fixed safety threshold, denoted as $\epsilon$, for model development. While this may hold in domains such as drug approvals or medical equipment (as illustrated by the authors' N95 mask example), applying a similar framework to AI models is more challenging and complex.
- (Minor) The model assumes that the test set used by regulators is drawn from the same distribution as the agent’s evaluation data. However, in the specific context of language models, techniques such as fine-tuning and reinforcement learning from human feedback (RLHF) can easily improve performance metrics if the evaluation distribution remains consistent. This weakens the argument that a single scalar value would sufficiently capture the intricacies of regulatory inspection.
- The authors propose a strictly increasing relationship between safety and cost, arguing that "safer models cost more to develop." However, they do not explicitly account for the trade-off between safety and the model's quality or usefulness in their framework. This omission raises questions, particularly since existing alignment approaches (e.g., RLHF) are often designed to balance helpfulness and harmlessness. In practice, a model could be made extremely safe (e.g., by providing only generic responses), but this could significantly reduce its usefulness without necessarily increasing development costs. In fact, under the authors' framework, one could submit a trivial model (e.g., one that always responds with "Thank you for your input"), bid the highest possible value, and meet the safety threshold $\epsilon$ to claim the regulator's compensation. This suggests that achieving safety in some cases may not necessarily be costly unless the model’s quality or usefulness is held constant.
- This issue could be exacerbated by the presence of open-source models like LLaMA, which may further incentivize the "gaming" of the regulatory system. Agents could enter the competition with low-cost variants of open-source models that prioritize safety at the expense of quality, primarily to secure the regulator’s compensation. Put it in a different way, low-quality models (which are safe but not useful) could flood the regulatory system, making it easier to claim compensation without delivering valuable AI products. This could distort incentives, where participants optimize for regulatory approval rather than producing high-quality, well-rounded models.

For the mechanism itself, a minor concern include the use of randomization, which introduces envy into the mechanism. With development costs potentially huge, this might lead to issues and discontent and distrust with the mechanism after the outcome is realized.

**Questions:**

Beyond the questions listed in the Weakness section, here are some additional questions I have:
- The framework assumes that the cost $M$ is the same across agents. This assumption seems unrealistic in practice, given that different agents may have varying models, training procedures, and resources, which makes the cost of aligning the safety levels different. If $M$ differs across agents, is there a way to adapt the framework to accommodate heterogeneous costs while maintaining its theoretical properties?
- The paper didn't mention incentive compatibility, a key issue in auction literature. Is truthful report of $b_i$ guaranteed?

---

> ### Author Response · Authors · 2024-11-15
> **Reviewer F4Wy Rebuttal (Part 1)**
>
> Thank you, Reviewer F4Wy, for your insightful review of our paper. We appreciate that you found our line of work novel, theoretically sound, and well-written. Below, we address all questions you raised.
>
> ## Weaknesses
>
> ---
>
> > **Weakness 1:** (Minor) The authors assume a fixed safety threshold, denoted as $\epsilon$, for model development. While this may hold in domains such as drug approvals or medical equipment (as illustrated by the authors' N95 mask example), applying a similar framework to AI models is more challenging and complex.
>
> **Response to Weakness 1:**
>
> - Various safety metrics exist that can already be applied to gauge AI model safety (*e.g.,* F1 Score, human-annotated error rate, win rate, or attack success rate for LLMs).
>
>
> While complex and challenging, quantifying the safety of AI models is still feasible and necessary. Using current safety evaluation metrics is better than the alternative: zero safety regulation on deployed AI models. Furthermore, our framework is general enough such that when an improved method for evaluating AI model safety arises, it can immediately be used for the model evaluation process.
>
>
> > **Weakness 2:** (Minor) The model assumes that the test set used by regulators is drawn from the same distribution as the agent’s evaluation data. However, in the specific context of language models, techniques such as fine-tuning and reinforcement learning from human feedback (RLHF) can easily improve performance metrics if the evaluation distribution remains consistent. This weakens the argument that a single scalar value would sufficiently capture the intricacies of regulatory inspection.
>
> **Response to Weakness 2:**
>
> - Techniques such as fine-tuning and RLHF add to agent costs (due to the need to collect more human/AI feedback and update the billions of parameters).
>
>
> We agree that fine-tuning and RLHF can improve performance metrics. However, performing fine-tuning and RLHF incurs added cost. This is exactly what we model within our paper: increased safety necessitates increased cost. We believe that there may be some confusion surrounding the safety-metric function $𝑆$ and its application in model evaluation. The inputs of $S$ are the model parameters $w$ and evaluation data $x$.
>
> - In the example provided by the reviewer, fine-tuning or RLHF would result in a new set of parameters $w'$ such that $S(w';x) = \epsilon' > S(w;x) = \epsilon$.
> - There is added cost for an agent to find $w'$ via fine-tuning or RLHF: $M(\epsilon') = p_{\epsilon'} > M(\epsilon) = p_{\epsilon}$.
>
>
> > **Weakness 3:** The authors propose a strictly increasing relationship between safety and cost, arguing that "safer models cost more to develop." However, they do not explicitly account for the trade-off between safety and the model's quality or usefulness in their framework. This omission raises questions, particularly since existing alignment approaches (e.g., RLHF) are often designed to balance helpfulness and harmlessness. In practice, a model could be made extremely safe (e.g., by providing only generic responses), but this could significantly reduce its usefulness without necessarily increasing development costs. In fact, under the authors' framework, one could submit a trivial model (e.g., one that always responds with "Thank you for your input"), bid the highest possible value, and meet the safety threshold  to claim the regulator's compensation. This suggests that achieving safety in some cases may not necessarily be costly unless the model’s quality or usefulness is held constant.
>
> **Response to Weakness 3:**
>
> - Our regulatory framework holds for all AI models and not just LLMs.
> - Our definition of safety is much more general than determining if the output of the model is "harmless" or not.
>
> The definition of safety that we use takes into account the usefulness of a model's output. For example, one may want to evaluate the F1 score of a model in order to ensure that it is minimizing the number of false positive and negative predictions (especially since false negative predictions can be very dangerous in healthcare settings). For LLMs, one may want to evaluate the attack success rate against submitted models on a benchmark such as [JailbreakBench](https://jailbreakbench.github.io) and/or assess the factuality of responses via human evaluation. As a result, an LLM that always responds "Thank you for your input" may be harmless, but it will fail to provide accurate responses and would be flagged for providing incorrect and unfactual responses by human evaluation. The evaluation metrics used to quantify safety are domain-specific and must incorporate evaluation of model quality. An active line of future research we are pursuing is determining which evaluations metrics are most effective across a variety of domains.

---

> ### Author Response · Authors · 2024-11-15
> **Reviewer F4Wy Rebuttal (Part 2)**
>
> > **Weakness 4:** This issue could be exacerbated by the presence of open-source models like LLaMA, which may further incentivize the "gaming" of the regulatory system. Agents could enter the competition with low-cost variants of open-source models that prioritize safety at the expense of quality, primarily to secure the regulator’s compensation. Put it in a different way, low-quality models (which are safe but not useful) could flood the regulatory system, making it easier to claim compensation without delivering valuable AI products. This could distort incentives, where participants optimize for regulatory approval rather than producing high-quality, well-rounded models.
>
> **Response to Weakness 4:**
>
> We believe that our response to Weakness 3 clarifies our definition of safety and provides an answer to this Weakness.
>
>
> > **Weakness 5:** For the mechanism itself, a minor concern include the use of randomization, which introduces envy into the mechanism. With development costs potentially huge, this might lead to issues and discontent and distrust with the mechanism after the outcome is realized.
>
> **Response to Weakness 5:**
>
> - Performing the randomization process multiple times reduces the likelihood of unfair outcomes.
>
> In practice, to avoid the possible unfair scenarios as detailed in the reviewer's question, we can repeat the randomization process $x$ times. For this to work, the regulator will store the number of times each agent has the higher safety bid $n_i$. Then, the regulator will award premium rewards to agents having value $n_i / x$ in the top half of all agents.
>
>
> ## Questions
>
> ---
>
> > **Question 1:** The framework assumes that the cost $M$ is the same across agents. This assumption seems unrealistic in practice, given that different agents may have varying models, training procedures, and resources, which makes the cost of aligning the safety levels different. If $M$ differs across agents, is there a way to adapt the framework to accommodate heterogeneous costs while maintaining its theoretical properties?
>
> **Response to Question 1:**
> - The motivation behind Assumption 2 is to generalize the relationship between safety and cost within the domain of AI regulation.
>
> In general, safer models cost more to develop. We agree that there may be some slight discrepancies between agents regarding this relationship in certain settings.
>
> - As the first paper proposing a mathematically-based regulatory framework to incentivize safer model deployment within the AI regulatory domain, we believe that cost function discrepancies are secondary to the primary concern of developing frameworks to tackle AI safety regulation.
>
> Allowing personalized $M_i$ functions is an active line of research we are conducting for follow-up work.
>
> > **Question 2:** The paper didn't mention incentive compatibility, a key issue in auction literature. Is truthful report of $b_i$ guaranteed?
>
> **Response to Question 2:**
>
> - Truthfulness of $b_i$ is not a major issue within our mechanism since it is verified by the regulator (auctioneer) itself.
> - Each agent must provide the regulator access to its model in order to verify its safety level.

---

### Official Review · Reviewer_LVmP · 2024-10-28

**Soundness:** 3
**Presentation:** 4
**Contribution:** 2
**Rating:** 5
**Confidence:** 3

**Summary:**

The paper addresses the challenges regulators face, particularly with the deployment of large language models that can amplify misinformation and societal division. It highlights the urgent need for effective regulatory frameworks to mitigate these risks and enhance user safety. Observing a gap in the availability of rigorous and realistic mathematical frameworks for AI regulation, the authors propose an innovative auction-based regulatory mechanism. This mechanism is designed to incentivize the development and deployment of safer AI models and encourage active participation in the regulatory process. It demonstrates through derived Nash Equilibria that the proposed auction mechanism effectively ensures that each participating agent’s optimal strategy aligns with submitting a model that exceeds a set minimum-safety threshold.

**Strengths:**

1. The topic considered in this paper is interesting and important. Regulations are needed to ensure AI safety.

2. Theoretical results are provided whose proofs can be found in the appendix. I didn't check all the mathematical proofs.

3. The paper is overall well-written and well motivated.

**Weaknesses:**

1. The way used by the paper to model the safety may not be realistic. It is assumed to be some safety level $s_i$ of a model $w_i$, which is expected to be less than $\epsilon$. How is the safety measured for AI models using the metric mapping $S$ in practice? For common foundation models and LLMs, it might be hard to evaluate $S$ for $w_i$, especially given the size of $w_i$. What if a model provider take advantage of the inaccuracy of the safety evaluation to benefit itself?

2. The proposed auction algorithm, together with the theoretical results and analysis seem quite standard. How does it differ from the classic all-pay auction results (for instance, Amann et al. 1996) in the setting for AI models? It is worth highlighting the technical novelty and emphasize why the proposed method is needed for AI models, given that it is claimed in Line 398-399 that "To the best of our knowledge there are no other comparable mechanisms for safety regulation in AI."

**Questions:**

1. What is the technical challenge in the considered auction problem for AI models, compared to classic auction problems?

2. Practical AI models are often very large. How can the safety of these model be evaluated? Given that the auction is done in a one shot setting, probably it is fine even if the model is large.

3. I am more concerned about the compensation $v_i^p$, which needs to be provided by a regulator to implement the proposed auction algorithm. Why is this practical for existing AI models? How large does the compensation need to be? According to bidding equilibrium in Theorem 2,  $v_i^p$ needs to be large for safer models. How could this be made up to compensate what the commercial AI models could achieve?

---

> ### Author Response · Authors · 2024-11-15
> **Reviewer LVmP Rebuttal (Weaknesses)**
>
> Thank you, Reviewer LVmP, for your insightful review of our paper. We appreciate that you found our work important, interesting, and well-written. Below, we address all questions you raised.
>
> ## Weaknesses
>
> ---
>
> > **Weakness 1:** The way used by the paper to model the safety may not be realistic. It is assumed to be some safety level $s_i$ of a model $w_i$, which is expected to be less than $\epsilon$. How is the safety measured for AI models using the metric mapping $S$ in practice? For common foundation models and LLMs, it might be hard to evaluate $S$ for $w_i$, especially given the size of $w_i$. What if a model provider take advantage of the inaccuracy of the safety evaluation to benefit itself?
>
> **Response to Weakness 1:**
>
> - The function $S$ is simply any metric that a regulator uses to gauge the safety performance $s_i$ of a model, represented by its parameters $w_i$ (*e.g.,* analyzing attack success rate on [JailbreakBench](https://jailbreakbench.github.io) for LLMs).
>
> We believe there may be some confusion regarding the function $𝑆$ and its application in model evaluation.  In SIRA, agents will send their models to the regulator, who will gauge their safety levels using $S$.
>
> - $S$ determines the safety level $s_i$ of a model $w_i$, but does not relate safety to cost.
>
> Confusion may have arisen within the relationship between safety levels $s_i$ and agent costs. Within our paper, we assume that safety level $s_i$ is related to cost, via function $M$, in an increasing manner (*i.e.,* a larger safety level comes with an increasingly large cost). Thus, agents that desire a larger safety level $s_i$, determined by $S$, will have to pay more to attain it.
>
> > **Weakness 2:** The proposed auction algorithm, together with the theoretical results and analysis seem quite standard. How does it differ from the classic all-pay auction results (for instance, Amann et al. 1996) in the setting for AI models? It is worth highlighting the technical novelty and emphasize why the proposed method is needed for AI models, given that it is claimed in Line 398-399 that "To the best of our knowledge there are no other comparable mechanisms for safety regulation in AI."
>
> **Response to Weakness 2:**
>
> - SIRA is specifically designed to mathematically formulate the AI regulation problem.
> - SIRA incorporates a reserve price (minimum safety bid required to win the deployment reward).
> - SIRA allocates multiple rewards to many ($n >> 2$) agents.
> - The derived agent utility (Equation 6) and derived equilibria in SIRA are novel and different than previous auction literature.
>
> We want to thank the reviewer for allowing us to clarify, and more clearly detail within our paper, the technical novelty of SIRA compared to other all-pay auction works. The setting of our paper versus previous all-pay auction literature is starkly different. In previous literature (Amann 1996 for instance), the equilibrium of a two-player asymmetric all-pay auction is determined. There is only one winner and one reward, and there is no floor that the players must bid over in order to win their reward. In contrast, SIRA is the first to formulate the AI regulatory process as an auction. Thus, SIRA must account for **(i)** many more agents, **(ii)** a required safety level for model deployment, and **(iii)** multiple rewards available to the participating agents. As a result, the agent utility function in SIRA is much different than those in previous all-pay auction literature. Therefore, our theoretical analysis in deriving an equilibrium given this new utility is novel. Finally, we prove that SIRA spurs increased bids compared to other baselines in this domain that we ourselves formulated (Reserve Thresholding, Section 4).

---

> ### Author Response · Authors · 2024-11-15
> **Reviewer LVmP Rebuttal (Questions)**
>
> ## Questions
>
> ---
>
> > **Question 1:** What is the technical challenge in the considered auction problem for AI models, compared to classic auction problems?
>
> **Response to Question 1:**
> We answer this question in Weakness 2 above.
>
> > **Question 2:** Practical AI models are often very large. How can the safety of these model be evaluated? Given that the auction is done in a one shot setting, probably it is fine even if the model is large.
>
> **Response to Question 2:**
> We answer this question in Weakness 1 above.
>
> > **Question 3:** I am more concerned about the compensation $v_i^p$, which needs to be provided by a regulator to implement the proposed auction algorithm. Why is this practical for existing AI models? How large does the compensation need to be? According to bidding equilibrium in Theorem 2, $v_i^p$ needs to be large for safer models. How could this be made up to compensate what the commercial AI models could achieve?
>
> **Response to Question 3:**
> - If a regulatory body and framework are established, all existing models would have to pass through them before continued use.
>
> Any existing models that do not meet the safety threshold would be barred from deployment (with the threat of governmental action).
>
> - Premium rewards provide incentive for agents that have existing models to make their model safer.
>
> For example, if the premium reward is a tax credit coupled with fast-tracked model deployment, Google, for example, may try to bid such a safe model that it is cleared to be deployed faster than one of its rivals, say OpenAI. In this way, the premium rewards still provide incentive for the agents of existing models to train them to be even safer.
>
> - The size of the premium reward depends upon the monetary limits of the regulator.
>
> The reviewer is correct that larger premium reward values $v_i^p$ correspond to the safer models submitted to the regulator (Theorem 2). As a result, the regulator should try to increase the value of its premium reward to be as large as possible. However, there is a limit to what regulators can offer agents. For example, regulators are not able to offer millions of dollars to each agent that builds a safe model. Thus, the value $v_i^p$ depends upon the monetary limits of the regulator.

---

> > ### Comment · Reviewer_LVmP · 2024-11-25
> >
> > Thank you for answering my questions.

---

> > > ### Author Response · Authors · 2024-11-25
> > > **Discussion Response**
> > >
> > > Dear Reviewer LVmP,
> > >
> > > Thank you for your response confirming that we have answered your questions. We are happy to address any remaining concerns if they exist. If all of your concerns have been addressed, we would request a reconsideration of your original score.

---

> > > > ### Author Response · Authors · 2024-12-02
> > > > **Reviewer Reply Deadline**
> > > >
> > > > Dear Reviewer LVmP,
> > > >
> > > > We sincerely appreciate your time and effort to review our work. With the deadline for discussion ending in less than 20 hours, we want to make sure that our responses have addressed all of your concerns. Any additional insight is very helpful for us. If all of your concerns have been addressed, we would request a reconsideration of your original score.
> > > >
> > > > Best,
> > > >
> > > > Authors

---

### Official Review · Reviewer_dJpW · 2024-10-31

**Soundness:** 2
**Presentation:** 3
**Contribution:** 2
**Rating:** 5
**Confidence:** 3

**Summary:**

This paper presents a new AI regulatory framework known as the Safety-Incentivized Regulatory Auction (SIRA), designed as an all-pay auction. SIRA aims to motivate model-building agents to prioritize safety beyond a minimum threshold by formulating the AI regulatory process as an asymmetric all-pay auction with incomplete information. In this framework, agents submit their models to a regulator, and those that meet or exceed a specified safety level become eligible for deployment and may also receive additional rewards based on the randomized pair comparison result. The authors theoretically prove that under some assumptions and when all agents adopt a certain strategy, the system reaches a Nash Equilibrium. Empirical results indicate that when safety threshold prices are in the middle (0.2~0.8), SIRA enhances safety compliance and agent participation by 20% and 15%, respectively compared with the basic regulatory method.

**Strengths:**

**Originality.** The approach presents a unique use of all-pay auction mechanisms in AI regulation, where each agent's utility is linked to model safety levels (training cost), model value (market returns), and premium (policy compensation), creating an incentive for improved safety compliance.

**Quality.** The paper theoretically derives Nash Equilibria to back the proposed incentive structure, demonstrating that agents' rational behavior leads them to exceed the minimum safety threshold. The experimental results align with the theoretical model.

**Clarity.** This paper is well-written and easy to follow. The authors provide clear descriptions of the auction-based model and detailed steps in the algorithmic design of SIRA, supported by both theoretical and empirical validation.

**Significance.** This paper tries to tackle an essential issue in AI regulation by encouraging safer model deployment.

**Weaknesses:**

**Rationality of the auction framework.** Considering the regulation process as an all-pay auction is not reasonable, at least in my opinion. Intuitively, safety-driven regulation establishes a minimum cost for the model-building agent. Every model-building agent must incur this cost, regardless of whether it can successfully meet the regulatory requirements. This represents an unavoidable exploration process within the model space. Even if we assume that all competitive agents know how to train their models to meet the safety threshold, accurately estimating the value of deployment remains a challenge. Thus, the framework may be overly simplistic in its approach to "safety" regulation.

**Feasibility of Assumptions 1 and 2.** Assumption 1 fails when a model-building agent maliciously injects backdoor triggers into the model by altering the training dataset. Assumption 2 is also not straightforward. More cost (e.g., computational resources) does not necessarily equate to better safety. Safety also depends on other factors, such as the learning paradigm, model architecture, loss function design, and hyperparameter selection.

**Performance at high thresholds.** As highlighted in the experiments, SIRA demonstrates limited advantages when safety thresholds approach the upper range (e.g., above 0.8), where its performance is similar to that of simpler reserve threshold models.

1. Evan Hubinger, et al., Sleeper Agents: Training Deceptive LLMs that Persist Through Safety Training, arXiv:2401.05566.

**Questions:**

**Q1.** Is there a reasonable mechanism for estimating the market value ($v_i^d$) of a model before it is submitted to the regulator or even before the training phase begins?

**Q2.** Considering that SIRA’s performance deteriorates at high safety thresholds, would a simple increase in the threshold serve as a better incentive in such cases, as it may more directly encourage safer model development?

**Q3.** The authors mention that safety evaluations rely on IID assumptions for both agent and regulator data. How would the proposed mechanism adapt to non-IID settings, where the agent's training data might be maliciously poisoned, or where the regulator's evaluation data is collected through other means?

**Q4.** Is the random comparison fair for all competitive agents? For example, if we have utility values such that $u_A > u_B > u_C > u_D$, and A and B are grouped together while C and D are grouped together, then B and D cannot receive the policy bonus. However, since $u_B > u_C$, this situation could be considered unfair to B.

---

> ### Author Response · Authors · 2024-11-15
> **Reviewer dJpW Rebuttal (Part 1)**
>
> Thank you, Reviewer dJpW, for your insightful review of our paper. We appreciate that you found our work original, clear, and significant. Below, we address all questions you raised.
>
> ## Weaknesses
>
> ---
>
> > **Weakness 1:** Rationality of the auction framework. Considering the regulation process as an all-pay auction is not reasonable, at least in my opinion. Intuitively, safety-driven regulation establishes a minimum cost for the model-building agent. Every model-building agent must incur this cost, regardless of whether it can successfully meet the regulatory requirements. This represents an unavoidable exploration process within the model space. Even if we assume that all competitive agents know how to train their models to meet the safety threshold, accurately estimating the value of deployment remains a challenge. Thus, the framework may be overly simplistic in its approach to "safety" regulation.
>
> **Response to Weakness 1:**
>
> - We believe it is rational that there exists a minimum cost incurred by each model-building agent in order to have its model deployed.
>
> This cost arises from placing effort into searching the model space for safe models. Simply put, **if a model is not safe enough to deploy, regardless of the cost incurred by the agent who built it, it should not be deployed.** As eloquently written in [Bengio et al. 2024]: "Safety cases are politically viable even when people disagree on how advanced AI will become, since it is easier to demonstrate a system is safe when its capabilities are limited. Governments are not passive recipients of safety cases: they set risk thresholds, codify best practices, employ experts and thirdparty auditors to assess safety cases and conduct independent model evaluations, and hold developers liable if their safety claims are later falsified."
>
> - Our proposed regulatory framework provides a guide for a regulatory body to incentivize safe model development and deployment.
>
> It is out of the scope of our work to detail how model-building agents incur the cost of safety training. In the case of LLMs, methods such as reinforcement learning from human feedback (RLHF) and fine-tuning allow agents to make their models safer.
>
> 1. Yoshua Bengio et. al. Managing extreme AI risks amid rapid progress, 2024.
>
> > **Weakness 2:** Feasibility of Assumptions 1 and 2. Assumption 1 fails when a model-building agent maliciously injects backdoor triggers into the model by altering the training dataset. Assumption 2 is also not straightforward. More cost (e.g., computational resources) does not necessarily equate to better safety. Safety also depends on other factors, such as the learning paradigm, model architecture, loss function design, and hyperparameter selection.
>
> **Response to Weakness 2:**
>
> - The regulator can use various defenses [Goldblum 2022; Zhao 2024; Gao 2020] to mitigate a wide variety of attacks, including backdoor attacks.
> - Defending against malicious attacks falls out of the scope of our proposed framework.
>
> **Remark:** The goal of our paper is to provide the first mathematically-based regulatory framework to incentivize safer model deployment within the AI regulatory domain. Defending against maliciously-submitted models is an interesting and important future line of work.
>
> - There is a cost associated with exploring various design factors, including the learning paradigm, model architecture, loss function design, and hyperparameter selection.
>
> In the examples provided by the reviewer, we agree that the learning paradigm, model architecture, loss function design, and hyperparameter selection do affect safety. However, there is a cost to investigate each of these provided examples. As a result, these all fall under "cost". Determining a relationship between cost and each one of these factors requires a detailed analysis that falls out of the scope of the paper.
>
> 2. Goldblum, Micah, et al. Dataset security for machine learning: Data poisoning, backdoor attacks, and defenses, 2022.
> 3. Zhao, Shuai, et al. A survey of backdoor attacks and defenses on large language models: Implications for security measures, 2024.
> 4. Gao, Yansong, et al. Backdoor attacks and countermeasures on deep learning: A comprehensive review, 2020.

---

> ### Author Response · Authors · 2024-11-15
> **Reviewer dJpW Rebuttal (Part 2)**
>
> > **Weakness 3:** Performance at high thresholds. As highlighted in the experiments, SIRA demonstrates limited advantages when safety thresholds approach the upper range (e.g., above 0.8), where its performance is similar to that of simpler reserve threshold models.
>
> **Response to Weakness 3:**
>
> - Reserve Thresholding (RT) is also a novel mechanism that we propose within our paper.
> - SIRA provably outperforms RT for bidding size and participation rate across *all* $\epsilon$ ranges.
> - SIRA is a more realistic and robust framework, as it can be used across various settings where the $\epsilon$ threshold can be vastly different.
>
>  While SIRA empirically demonstrates limited advantages at the upper range of safety thresholds versus RT, it still provably improves the bidding size compared to RT (albeit small). Below, we compare the participation rate and bidding size between SIRA and RT (in the Uniform setting).
>
> | $\epsilon$ Range | SIRA Participation | SIRA Bid | RT Participation | RT Bid  |
> | -------- | ------ | -------- | -------- | --------|
> | (0, 0.2)     | **86.273%**     |  **0.145**    | 85.352% | 0.105|
> | (0.2, 0.4)     | **61.788%**     | **0.358**  | 57.640%  | 0.305 |
> | (0.4, 0.6)     | **38.414%**     | **0.567**     | 30.402% |  0.505|
> | (0.6, 0.8)     | **15.514%**     | **0.753**     | 10.297% | 0.705 |
> | (0.8, 1.0)     | **1.633%**     | **0.903**     | 1.424% | 0.900 |
>
> As expected from the results of our theory (Section 5), SIRA outperforms RT in participation rate and bid size across all $\epsilon$ ranges.
>
> ## Questions
>
> ---
>
> > **Question 1:** Is there a reasonable mechanism for estimating the market value ($v_i^d$) of a model before it is submitted to the regulator or even before the training phase begins?
>
> **Response to Question 1:**
> - In Auction literature [Amann 1996; Bhaskar 2018; Tardos 2017], agent valuations are private and arise from nature; no mechanism estimates model deployment value.
> - It is realistic for many companies to place a market value on its own intellectual property and products.
>
> In practice, these valuations are determined in house. For example, Google may perform market research to determine the value (revenue generation) of a model, like Gemini, before it is released.
>
> > **Question 2:** Considering that SIRA’s performance deteriorates at high safety thresholds, would a simple increase in the threshold serve as a better incentive in such cases, as it may more directly encourage safer model development?
>
> **Response to Question 2:**
>
> - Increasing the threshold only discourages agents with lower total value $V_i$ from participating.
>
> As one can see in Figures 2 & 3, raising the threshold results in lower participation (while the bids increase in size).
>
> - A straightforward method to incentivize safer model deployment would be to increase the premium reward.
>
> Increasing the premium reward would shift the probability mass of total value $V_i$ towards 1 for agents. Consequently, more agents would have values closer to 1, which results in more agents willing to train a model that is able to clear the higher safety threshold.
>
> > **Question 3:** The authors mention that safety evaluations rely on IID assumptions for both agent and regulator data. How would the proposed mechanism adapt to non-IID settings, where the agent's training data might be maliciously poisoned, or where the regulator's evaluation data is collected through other means?
>
> **Response to Question 3:**
> - As detailed in our Future Work (Section 7), one possible solution is the requirement that data must be shared (in a private and anonymous manner) between each agent and the regulator.
> - Another possible solution would be the regulator collecting more data on its own (with possible assistance from agents).
> - The regulator can employ various defenses to mitigate malicious attacks (see response to Weakness 2).
>
> > **Question 4:** Is the random comparison fair for all competitive agents? For example, if we have utility values such that $u_A > u_B > u_C > u_D$, and A and B are grouped together while C and D are grouped together, then B and D cannot receive the policy bonus. However, since $u_B > u_C$, this situation could be considered unfair to B.
>
> **Response to Question 4:**
>
> - Performing the randomization process multiple times reduces the likelihood of unfair outcomes.
>
> In practice, to avoid the possible unfair scenarios as detailed in the reviewer's question, we can repeat the randomization process $x$ times. For this to work, the regulator will store the number of times each agent has the higher safety bid $n_i$. Then, the regulator will award premium rewards to agents having value $n_i / x$ in the top half of all agents.

---

> ### Comment · Reviewer_dJpW · 2024-11-23
>
> Thanks to the authors for their response. However, the response to Weaknesses 1 & 2 cannot fully address my concern.
>
> **Weakness 1 on the rationality of the auction framework**. What the authors claim, i.e., "there exists a minimum cost incurred by each model-building agent in order to have its model deployed," is exactly what I pointed out: "Every model-building agent must incur this cost, regardless of whether it can successfully meet the regulatory requirements." This claim, in my point of view, weakens the rationality of the auction framework.
>
> To clarify my point, in an auction, one can choose not to bid for a certain item, and thus, there can be no cost for them. However, in the context of regulating LLMs, every model-building agent has to pay for the cost of pre-/post-training an LLM to improve the task performance and meet the underlying safety constraints once it starts to develop any LLM. Thus, there is an essential difference between audition and regulation.
>
> **Weakness 2 on Feasibility of Assumptions 1 and 2**. Actually, this weakness has also been mentioned by Reviewer dJpW and F4Wy. This confirms my concern on the feasibility of Assumptions 1 and 2. Especially, the discussion on the "cost" definition cannot support Assumption 2 that there exists a strictly increasing function M that maps safety to cost. **This assumption is too strong and not practical**. Although determining a relationship between cost and each one of these factors falls out of the scope of the paper, **the relationship between safety and cost cannot be oversimplified as a strictly increasing function**. What's more, the safety score and the cost score themselves are not even easy to quantify as a scalar in practice since both safety constraints and cost involve many factors, as agreed by the authors.

---

> ### Comment · Reviewer_dJpW · 2024-11-23
>
> Thanks for the authors' detailed answers to my concerns and questions.  Q2, Q3, and Q4 have been well addressed.
>
> For Weakness 3,
>
> 1) Although reserve thresholding (RT) is also a novel mechanism that the author proposes, the performance of SIRA at high thresholds compared with the baseline RT weakens the main contribution of SIRA, which occupies three pages of the main body.
> 2) Though SIRA outperforms RT when $\epsilon<0.8$, setting $\epsilon<0.8$ may not be meaningful since $\epsilon<0.8$ may not be acceptable by the market, laws, or politics. In other words, there is an implicit constraint for the epsilon. To justify the strength of SIRA when $\epsilon<0.8$, the author should justify the reasonable range of $epsilon$. It is better to involve current LLMs and current safety constraints to justify the range of $epsilon$.
>
> For Question 1, as answered by the authors, it seems to be challenging to develop a standard mechanism to estimate model deployment value. This challenge makes SIRA impractical.

---

> > ### Author Response · Authors · 2024-11-24
> > **Discussion Response (Part 1)**
> >
> > We apologize if our earlier response was unable to address your concerns. Thank you for getting back to us, and we take this opportunity to clarify them further.
> >
> > > **Weakness 1 on the rationality of the auction framework.** What the authors claim, i.e., "there exists a minimum cost incurred by each model-building agent in order to have its model deployed," is exactly what I pointed out: "Every model-building agent must incur this cost, regardless of whether it can successfully meet the regulatory requirements." This claim, in my point of view, weakens the rationality of the auction framework.
> >
> > **Response:**
> >
> > - The cost incurred by each agent does not only consider pre- or post-training of a model.
> > - Standard model training also affects safety performance (*e.g.,* a well-trained cancer-classifying model will achieve a better F1-Score than an untrained model).
> >
> > Agents that train a model inherently incur a cost towards improving safety, even if it is small.
> >
> > >To clarify my point, in an auction, one can choose not to bid for a certain item, and thus, there can be no cost for them. However, in the context of regulating LLMs, every model-building agent has to pay for the cost of pre-/post-training an LLM to improve the task performance and meet the underlying safety constraints once it starts to develop any LLM. Thus, there is an essential difference between auction and regulation.
> >
> > - In our proposed framework, agents are allowed not to participate, thereby not biding, and will incur no cost as a result. Thus, our framework indeed aligns with that of an auction.
> >
> > Like an auction, agents that wish not to participate (and thus do not bid) will not incur any cost.
> >
> > > **Weakness 2 on Feasibility of Assumptions 1 and 2.** Actually, this weakness has also been mentioned by Reviewer F4Wy. This confirms my concern on the feasibility of Assumptions 1 and 2. Especially, the discussion on the "cost" definition cannot support Assumption 2 that there exists a strictly increasing function M that maps safety to cost. This assumption is too strong and not practical. Although determining a relationship between cost and each one of these factors falls out of the scope of the paper, the relationship between safety and cost cannot be oversimplified as a strictly increasing function.
> >
> > **Response:**
> >
> > - Providing the first theoretically-backed guarantees for AI safety regulation required the construction of new assumptions within the regulatory setting.
> > - While general, our assumption is realistic as **it does not make sense to be able to achieve more safety with less cost**.
> >
> > We agree that the additions of assumptions act as limitations towards the realism of any theoretical approach. We argue, however, that the first papers in unexplored research areas often include assumptions in order to begin proposing theory-backed solutions towards solving the problem at hand. Furthermore, we do not believe that the assumption between cost and safety laid out in our paper is unrealistic. In general, spending more to train a model (including pre- and post-training) will result in greater safety. In an area with little to no literature, this assumption is a general yet realistic insight into the empirical relationship between cost and safety.
> >
> > - Our goal is to spur future research into the regulatory AI domain that will chip away at the strength of the assumptions utilized.
> >
> > As detailed above, Assumption 2 generally models the relationship between cost and safety, and provides an avenue towards analysis. We believe that, as a first step towards tackling AI regulation, this is a reasonable assumption. Making our assumptions more realistic is a valid scope of future research.
> >
> >
> > > **Comment:** What's more, the safety score and the cost score themselves are not even easy to quantify as a scalar in practice since both safety constraints and cost involve many factors, as agreed by the authors.
> >
> > **Response:**
> >
> > We believe there is a misunderstanding on these aspects, and we respectfully disagree with the reviewer on this point. For instance, within all of the LLM safety alignment literature, even OpenAI [Ouyang 2022, Christiano 2017], a scalar valued reward is used to ensure that a model is safety aligned  [Kaufmann 2023]. While we agree that the cost involves many factors, it is reasonable to be able to estimate the monetary value of each cost (*e.g.,* the cost to collect data for RLHF or the cost for more compute time). As a result, cost can be reflected entirely in monetary value, which is a scalar value.
> >
> > 1. Ouyang, Long, et al. "Training language models to follow instructions with human feedback." Advances in neural information processing systems 35, 2022.
> > 2. Christiano, Paul F., et al. "Deep reinforcement learning from human preferences." Advances in neural information processing systems 30, 2017.
> > 3. Kaufmann, Timo, et al. "A survey of reinforcement learning from human feedback.", 2023.

---

> ### Author Response · Authors · 2024-11-24
> **Discussion Response (Part 2)**
>
> > **Comment:** Although reserve thresholding (RT) is also a novel mechanism that the author proposes, the performance of SIRA at high thresholds compared with the baseline RT weakens the main contribution of SIRA, which occupies three pages of the main body.
>
> **Response:**
>
> - We reiterate that SIRA is still provably better than reserve thresholding (RT) across the board.
> - SIRA is only slightly better than RT at high thresholds in the experiments we run; **SIRA may greatly outperform RT at high thresholds in other scenarios**.
>
> **We believe that it is unfair to penalize our work for proposing two novel frameworks in a research area that currently has zero alternatives.** Furthermore, SIRA is theoretically guaranteed to outperform RT in every instance. This is valuable, and would result in SIRA being implemented over RT in all real-world settings. Finally, the close performance at high thresholds arises experimentally due to the distributions used for total value $V_i$. **In practice, this gap would be much larger if more desirable rewards are provided by the regulator**. We detailed this in our original rebuttal, in response to Q2.
>
> > **Comment:** Though SIRA outperforms RT when $\epsilon < 0.8$, setting $\epsilon < 0.8$ may not be meaningful since $\epsilon < 0.8$ may not be acceptable by the market, laws, or politics. In other words, there is an implicit constraint for the $\epsilon$. To justify the strength of SIRA when $\epsilon < 0.8$, the author should justify the reasonable range of $\epsilon$. It is better to involve current LLMs and current safety constraints to justify the range of $\epsilon$.
>
> **Response:**
>
> We believe that our response to the comment above addresses this comment. In summary, SIRA always outperforms RT so there is never a reason to use RT over SIRA. Second, in many realistic scenarios, it may be the case that the total value is quite large for all agents and is not reflective of the distributions used within our experiments. In this case, SIRA would improve further at the larger thresholds.
>
> Finally, we still provide an example of settings where $\epsilon < 0.8$ is applicable in law. When taking the Universal Bar Exam (UBE) to become certified to practice law, **the most stringent states require a score of 270 out of a possible 400, a 67.5% score** ($\epsilon=0.675$). Many other examinations to ensure safe professional expertise (*e.g.*, Step 1/2 tests in Medicine, or Professional Engineering exams) require passage rates much lower than 80%. As a result, there are many instances where safe passage does not require large epsilon values.
>
> > **Comment:** For Question 1, as answered by the authors, it seems to be challenging to develop a standard mechanism to estimate model deployment value. This challenge makes SIRA impractical.
>
> **Response:**
>
> - SIRA does not need to estimate the value the model deployment of any agent.
>
> We believe there is some confusion surrounding model deployment value. Simply, model deployment value is a *agent-specific* value that each agent has internally (*e.g.,* revenue generation or market share percentage). For example, [OpenAI estimates that ChatGPT will bring in 2.7 billion dollars in revenue this year](https://www.nytimes.com/2024/09/27/technology/openai-chatgpt-investors-funding.html). Within the analysis of SIRA, akin to analysis of auctions, we provide theoretical results for any distribution of deployment value *across all agents* (Theorem 2). We then provide explicit equilibria for two given distributions (Corollaries 1 and 2). As an example, in the case of a Uniform distribution (Corollary 1), we provide an equilibrium in scenarios where it is equally likely that a random agent has an extremely large model deployment value as it does a small value.

---

> ### Comment · Reviewer_dJpW · 2024-11-25
>
> I really appreciate the authors for their prompt and thorough response. I believe that developing a theoretical framework for the AI regulation system is both important and promising. However, I still have concerns about the rationality of the auction system and the assumptions within the current framework. Addressing these issues would strengthen the paper.
>
> Regarding the rationality of the auction framework, the cost of developing VLMs for model-making agents is unavoidable, even if the agents temporarily choose not to submit their models. This is because agents must always submit their models to the regulator's black-box evaluation system to determine whether their VLMs meet the safety threshold.
>
> As for Assumption 2, I believe the relationship between safety and cost is oversimplified. For example, in adversarial machine learning, a better regulator can enhance the adversarial robustness of a neural network without increasing computing resources. Although the authors attempt to broaden the definition of "cost," this expansion makes quantifying the "cost" more challenging and ultimately undermines the practicality of the paper.
>
> Additionally, I've noticed that some other reviewers share concerns about the assumptions and the trade-off between safety and usefulness. As a result, I have decided to increase my score to 5 with a middle confidence level of 3.

---

> > ### Author Response · Authors · 2024-11-25
> > **Discussion Response (Part 3)**
> >
> > Thank you for the continued discussion.
> >
> > > **Comment:** Regarding the rationality of the auction framework, the cost of developing VLMs for model-making agents is unavoidable, even if the agents temporarily choose not to submit their models. This is because agents must always submit their models to the regulator's black-box evaluation system to determine whether their VLMs meet the safety threshold.
> >
> > **Response:**
> >
> > Thank you for this point. We still believe that there is some confusion surrounding the requirement of cost in our auction formulation. **As shown in our mathematical formulation in Equation 6, an agent who decides to either (a) not bid or (b) bid an unsafe model ($b_i = 0$) will incur zero cost and maintain a utility of zero.**
> >
> > The regulatory paradigm that we envision, which mirrors current regulatory systems (*e.g.,* FDA, FAA, etc.), is one in which no models are placed onto the market unless they have been fully vetted and regulated. **Agents must pay the price to bring a safe model to market.** All model-building agents understand, prior to development, that they will have to clear the regulatory system before their models are available for public use. Thus, agents weigh whether it is in their best interest to participate and develop a model or not. We see this theoretically in Corollaries 1 & 2, namely Equations 10 and 13. If, given a total value $V_i$, an agent's utility is not positive, it will not build a model. **This is reflected in markets today. A person will not enter the market for making house paint if they know that the costs for producing paint with the minimal lead content requirement are too large for them to turn a profit**.
> >
> > > **Comment:** As for Assumption 2, I believe the relationship between safety and cost is oversimplified. For example, in adversarial machine learning, a better regulator can enhance the adversarial robustness of a neural network without increasing computing resources.
> >
> > **Response:**
> >
> >
> > In the case of adversarial robustness, for example, there is a cost to determining what type of optimization method works best for your data (*e.g.,* Fast Gradient Sign Method or Projected Gradient Descent). In general, to improve the safety of a model, agents must incur extra cost to scope out many factors. These include the methods originally detailed by the reviewer: learning paradigm, model architecture, loss function design, and hyperparameter selection.
> >
> >
> > > **Comment:** Although the authors attempt to broaden the definition of "cost," this expansion makes quantifying the "cost" more challenging and ultimately undermines the practicality of the paper.
> >
> >
> > **Response:**
> >
> > We respectfully disagree that we are broadening the definition of "cost". As we state within our paper (Lines 165-167): *"The assumption that a strictly increasing function M maps safety to cost is realistic, because achieving higher safety levels typically requires greater resources. Safer models often demand more data, advanced tuning, and extensive validation, all of which increase costs".*
> >
> >
> > - We do not limit the scope of what cost is to only pre- or post-training.
> > - We have consistently detailed that "costs" towards improving model safety can arise from various avenues and investigations (shown from our quote above).
> > - The total cost of these investigations can be quantified as a monetary value (which is practical).
> >
> > The area of theoretically-backed frameworks for AI regulation is exceptionally sparse. There are no previous frameworks, let alone assumptions or theory, to build on. **Our work is the first to establish theoretical results and assumptions in this area.** While we agree that the additions of assumptions act as limitations towards the realism of any theoretical approach, *it is unreasonable to believe that the very first theory-backed solution in a research area will solve the entire problem with no assumptions utilized*. Furthermore, we do not believe that Assumption 2 is unrealistic, as it models the generic relationship between safety and cost.
> >
> > **Remark:** We would like to take this opportunity to emphasize the core contribution of our work. Our goal is to propose the first theory-backed AI regulatory framework that incentivizes safer model development and deployment. We believe that our paper takes a big stride towards implementable regulatory AI frameworks. With such a difficult and complex problem, it is nearly impossible to solve in its entirety in one shot. We hope that our paper will spur future research into this area and soon provide a robust solution for governments to implement.

---

> > > ### Author Response · Authors · 2024-12-02
> > > **Reviewer Reply Deadline**
> > >
> > > Dear Reviewer dJpW,
> > >
> > > We sincerely appreciate your time and effort to review our work. With the deadline for discussion ending in less than 20 hours, we want to make sure that our responses have addressed all of your concerns. Any additional insight is very helpful for us. If all of your concerns have been addressed, we would request a reconsideration of your current score.
> > >
> > > Best,
> > >
> > > Authors

---

### Official Review · Reviewer_Ft8N · 2024-10-31

**Soundness:** 3
**Presentation:** 3
**Contribution:** 3
**Rating:** 6
**Confidence:** 3

**Summary:**

This paper proposes a novel framework of auction-based regulatory mechanism as an asymmetric and incomplete all-pay auction. The mechanism is described mathematically and also shows good empirical results of enhancing safety and participation rates. The framework consists of a regulator and multiple participating agents. Overall, this is an interesting framework with good potentials to explore and create safer and more robust AI regulatory.

**Strengths:**

The paper is well-written and well-supported by both theoretical proofs and empirical results. It addresses the important area of AI regulatory via a multi-agent economic, game-theory type framework. There are a few assumptions to simplify the mechanism but they appear to be acceptable/realistic such as i) the regulator and the participating agents use data from the same distribution to evaluate and submit the safety level, and ii) safer models cost more to develop. These assumptions perhaps need more clarification/grounding or adjustment to become more applicable and feasible in practice. A safer model can tend to cost more to develop but perhaps cost and safety might not always be strictly increasing. The paper has help enhance the current AI regulatory work with a well-formulated framework and has a potential to have some significance in this domain.

**Weaknesses:**

While the paper is well-supported in the mathematical formulation and proofs, it perhaps could have provided more evidence on the experiments and empirical data. More description of how this framework can be applied in AI regulatory or in practice might help ground it further and make it relevant to a wider group of audiences.

**Questions:**

* What is the rationale of choosing the Beta and Uniform distribution (beyond what is described in line 323-324). Are there any related works that you could cite to support this choice of distributions?

* What is the scaling of complexity and cost (such as evaluation and communication) as the number of the agents increase? Are there any risks of agents colluding to achieve a suboptimal safety level?

---

> ### Author Response · Authors · 2024-11-15
> **Reviewer Ft8N Rebuttal**
>
> Thank you, Reviewer Ft8N, for your insightful review of our paper. We appreciate that you found our work well-written, well-supported, and can help "enhance the current AI regulatory work with a well-formulated framework and has a potential to have some significance in this domain". Below, we address all questions you raised.
>
> ## Weaknesses
>
> ---
>
> > **Weakness 1:** While the paper is well-supported in the mathematical formulation and proofs, it perhaps could have provided more evidence on the experiments and empirical data.
>
> **Response to Weakness 1:**
> - We have provided an additional ablation study within our Global Response that affirms the increasing relationship between safety and cost.
>
>
> > **Weakness 2:** More description of how this framework can be applied in AI regulatory or in practice might help ground it further and make it relevant to a wider group of audiences.
>
> **Response to Weakness 2:**
>
> - The goal of this paper is to introduce a mathematically-based regulatory framework for incentivizing safer AI model deployment and detail (prove) its theoretical guarantees.
> - We dive into certain practical applications within Appendix D, namely extending SIRA to repeated regulatory auctions (which is realistic in practice).
> - We are working on a follow-up report that details how our framework can be applied in practice.
>
> We believe, as the reviewer mentions, that our paper will be a launchpad to begin to "explore and create safer and more robust AI regulatory" frameworks. As a first step, we aimed to provide the theoretical backing of such a framework. In parallel, we are working on a policy-based report to implement an AI regulatory framework such as our own in practice. This report will focus more on the specific details surrounding implementation and less about the mathematical guarantees of SIRA.
>
> ## Questions
>
> ---
>
> > **Question 1:** What is the rationale of choosing the Beta and Uniform distribution (beyond what is described in line 323-324). Are there any related works that you could cite to support this choice of distributions?
>
> **Response to Question 1:**
> - Uniform distributions are commonly utilized to analyze all-pay auctions (as detailed in Lines 299-301) [Amann 1996; Bhaskar 2018; Tardos 2017].
> - We were interested in analyzing more than just a Uniform distribution (which is the usual choice for all-pay analysis), and the Beta distribution seems like a realistic choice in certain settings (as detailed in Lines 323-324).
>
>
> > **Question 2:** What is the scaling of complexity and cost (such as evaluation and communication) as the number of the agents increase? Are there any risks of agents colluding to achieve a suboptimal safety level?
>
> **Response to Question 2:**
> - Complexity and cost depends upon the size and bandwidth of the regulator.
>
> There are examples of regulatory bodies that regulate a large number of products in a reasonable amount of time. For example, the FDA oversees approximately 2.1 Trillion dollars worth of food, tobacco, and medical products ([per its own numbers](https://www.fda.gov/media/168049/download)). Furthermore, the FDA has four approaches to speed-up the regulatory process for drug approval: Priority Review, Breakthrough Therapy, Accelerated Approval, and Fast Track. That being said, budget cuts and a lack of resources can limit the number of products reviewed, and increase the review process length.
>
> - SIRA scales linearly if there are enough resources.
>
> As long as there are enough people and resources to review submitted models, each submitted model can be analyzed by one regulatory agent.

---

> ### Comment · Reviewer_Ft8N · 2024-11-22
>
> Thanks for your responses which clarified my questions. However, I concur to other reviewers' points about the practical aspects of Assumption 1 and 2, and the rationality of the auction framework. The paper still has a good potential to contribute to the domain. I will retain the recommendation towards accept but have reduced the score to Weak accept to reflect the concerns about the feasibility of the assumptions.

---

> > ### Author Response · Authors · 2024-11-25
> > **Discussion Response**
> >
> > Thank you for your response, and we appreciate that you find our paper has "good potential to contribute to the domain" of AI regulatory frameworks. Below, we address your comment.
> >
> > > **Comment:** I concur to other reviewers' points about the practical aspects of Assumption 1 and 2, and the rationality of the auction framework.
> >
> > **Response:**
> >
> >
> > - The area of theoretically-backed frameworks for AI regulation is exceptionally sparse; there are no previous frameworks, let alone assumptions or theory, to build on.
> > - Our work is the first to establish theoretical results and assumptions in this area.
> > - While general, **Assumption 2 is realistic as it does not make sense to be able to achieve more safety with less cost**.
> >
> > While we agree that the additions of assumptions act as limitations towards the realism of any theoretical approach, it is unreasonable to believe that the very first theory-backed solution in a research area will solve the entire problem with no assumptions utilized. Furthermore, we believe that Assumption 2 is reasonably realistic to start research in the domain of AI regulation, as it models the generic relationship between safety and cost.
> >
> > **Remark:** We would like to take this opportunity to emphasize the core contribution of our work. Our goal is to propose the first theory-backed AI regulatory framework that incentivizes safer model development and deployment. We believe that our paper takes a big stride towards implementable regulatory AI frameworks. With such a difficult and complex problem, it is nearly impossible to solve in its entirety in one shot. We hope that our paper will spur future research into this area and soon provide a robust solution for governments to implement.

---

### Author Response · Authors · 2024-11-15
**Global Response**

Thank you to all the reviewers for their reviewing service and paper feedback. We are happy to see that the reviewers agree that we are working on an important research problem, that our paper is well-written, and that our theory-based approach is novel and promising.

Below, we provide additional empirical results that affirm our stated relationship between safety and cost on real-world data.

## Ablation Study

---

- We conduct an ablation study to demonstrate that in realistic settings, safety is mapped to cost in a monotonically increasing way (as detailed in Assumption 2).

While there are many factors to consider when gauging safe AI deployment, we analyze model fairness, via equalized odds, for image classification in this study. Equalized odds measures if different groups have similar true positive rates and false positive rates (lower is better).

- We train VGG-16 models on the Fairface dataset [Kärkkäinen 2019] for 50 epochs (repeated ten times with different random seeds), and consider a gender classification task with race as the sensitive attribute.

Models with the largest validation classification accuracy during training are selected for testing.

- Many types of costs exist for training safer models, such as extensive architecture and hyper-parameter search. In this study, we consider the cost of an agent acquiring more minority class data.

This leads to a larger and more balanced dataset. We simulate various mixtures of training data, starting from a 95:5 skew and scaling up to fully balanced training data with respect to the sensitive attribute. In our study, we gauge equalized odds performance on well-balanced test data for the models trained on various mixtures of data. Below we tabulate our results.

| Minority Class % | Mean Equalized Odds Score |
| -------- | -------- |
| 5%     | 22.55     |
| 10%     | 22.31     |
| 15%     | 18.97     |
| 20%     | 17.46     |
| 25%     | 15.78    |
| 30%     | 15.44     |
| 35%     | 13.09     |
| 40%     | 11.01     |
| 45%     | 9.83     |
| 50%     | 9.38     |

- The equalized odds score decreases (the model becomes safer) when collecting more minority class data (increased cost).

To adjust equalized odds to fit into the setting where $\epsilon \in (0,1)$, one can invert and normalize the equalized odds score. We will upload a new version of our paper that includes this ablation study (with a scatter plot of the relationship between safety and cost shown in the table above).

1. Kärkkäinen et. al, Fairface: Face attribute dataset for balanced race, gender, and age, 2019.

## Contribution

---

We want to emphasize the importance of furthering research into AI regulatory frameworks. The deployment and usage of AI models is often unchecked. Lax regulation of AI deployment has led to, and may further accelerate in the future, the proliferation of misinformation and harmful effects on society. We believe that our paper takes a big step towards a valuable societal goal of establishing an effective and mathematically-backed regulatory framework that governing bodies can implement. In summary, our paper contributes a novel mechanism to the area of AI regulation that:

1. Formulates the regulatory process realistically as an auction, where there is one regulating body and many model-building agents.
2. Leverages auction theory to derive equilibria such that rational agents are incentivized to both participate in the regulatory process and submit safer models.
3. Empirically improves model safety by over 20% and participation rates by 15% compared to baseline regulatory mechanisms.

---

### Author Response · Authors · 2024-11-21
**Request for Continued Discussion**

Dear Reviewers,

We want to thank you again for your reviewing service. We believe that our rebuttals have answered the questions raised within your reviews. If so, confirmation that we have indeed answered your concerns would be appreciated. If not, we are happy to continue the discussion before the discussion phase ends in a few days.

Best,
Authors

---

### Author Response · Authors · 2024-11-29
**Update and Followup**

Dear Reviewers,

Thank you for your engagement and discussion.

As an update, we have added our ablation study, including the promised figure, within our revised paper (in Appendix C.1). After the revision deadline, we also added a section exploring and explaining Assumption 2. Namely, we detail the motivation and generality of the assumption, pointing towards our new Ablation Study as evidence. Likewise, we describe that, in a space with no assumptions let alone theoretical analysis, we use Assumption 2 to establish the first theoretical results for safety-incentivized AI regulatory frameworks. We note that our future research aims to further relax the assumptions made within our paper.

With the discussion period ending this Monday, we also wanted to make sure that we have clarified all questions and concerns. If not, we are happy to clarify any questions before the deadline. If all of your concerns have been addressed, we would request a reconsideration of your scores.

Best,

Authors

---

### Meta-Review · Area_Chair_tfzg · 2024-12-20

**Metareview:**

This paper looks at the problem of regulating AI models, specifically for safety. This is approached through theoretical framework, through an all-pay auction with companies and a regulator. The authors find Nash equilibria and have theoretical results.

Reviewers agree that this is a very important problem, and that the approach taken is novel and interesting. Reviewers also agree that the paper is well-written, which I agree with. I agree that these are all key strengths of the paper.

Unfortunately, all reviewers agree on a key limitation: the restrictiveness of the assumptions, specifically Assumption 2. There is of course a fine line between have realistic assumptions and those that allow for the kind of theory this paper does. The authors have defended their assumptions in rebuttal and with additional text in the paper (and an ablation study). Upon further discussion with reviewers, however, we all find that this is still a key limiting factor of this paper. I encourage the authors to take these concerns into account for a future version of the paper, actively acknowledging and tackling these issues even earlier in the paper (eg by giving specific examples of where Assumption 2 is both realistic *and not realistic*).

**Additional Comments On Reviewer Discussion:**

The authors provided detailed rebuttal and changes during the discussion period. Most of the reviewers' concerns were addressed, which is good. However, all reviewers were unconvinced by the authors' response/claims about the limitations of Assumption 2, a key assumption in the analysis. Reviewer Ft8N reduced their score (to 6), and other reviewers did not increase their score above 5, all due to this.

Reviewer dJpW also specifically has concerns about ignoring the cost of training VLMs, which I agree is important, but I think of smaller concern.

---

### Decision · Program_Chairs · 2025-01-22

Reject